# PEARL: Parallel Speculative Decoding with Adaptive Draft Length

**Tianyu Liu**[1,2*]  **Yun Li**[3†]  **Qitan Lv**[1]  **Kai Liu**[3]  **Jianchen Zhu**[3]  **Winston Hu**[3]  **Xiao Sun**[2†]
[1]University of Science and Technology of China
[2]Shanghai AI Laboratory
[3]Tencent
`{tianyu_liu, qitanlv}@mail.ustc.edu.cn`
`yunli.charles@gmail.com`
`sunxiao@pjlab.org.cn`

## Abstract

Speculative decoding (SD), where an extra draft model is employed to provide multiple *draft* tokens first, and then the original target model verifies these tokens in parallel, has shown great power for LLM inference acceleration. However, existing SD methods suffer from the mutual waiting problem, i.e., the target model gets stuck when the draft model is *guessing* tokens, and vice versa. This problem is directly incurred by the asynchronous execution of the draft model and the target model and is exacerbated due to the fixed draft length in speculative decoding. To address these challenges, we propose a conceptually simple, flexible, and general framework to boost speculative decoding, namely **P**arallel sp**E**culative decoding with **A**daptive d**R**aft **L**ength (PEARL). Specifically, PEARL proposes *pre-verify* to verify the first draft token in advance during the drafting phase, and *post-verify* to generate more draft tokens during the verification phase. PEARL parallels the drafting phase and the verification phase via applying the two strategies, and achieves adaptive draft length for different scenarios, which effectively alleviates the mutual waiting problem. Experiments on various text generation benchmarks demonstrate the effectiveness of our PEARL, leading to a superior speed up performance up to **4.43**× and **1.50**×, compared to auto-regressive decoding and vanilla speculative decoding, respectively. Our code is available at `https://github.com/smart-lty/ParallelSpeculativeDecoding`.

## 1 Introduction

Large language models (LLMs) such as GPT-4, LlaMA, and DeepSeek (Achiam et al., 2023; DeepSeek-AI, 2024; Bommasani et al., 2021; Touvron et al., 2023) have dominated natural language understanding and generation (Khurana et al., 2023) over a wide range of applications. However, the substantial inference latency of these LLMs has emerged as a significant obstacle bounding their broader application in scenarios with restricted computational resources. This latency primarily originates from the auto-regressive token-by-token decoding process wherein decoding $K$ tokens requires $K$ serial runs of LLMs, incurring exacerbated latency with both the length of generated tokens and the model scale.

To address this challenge, extensive research efforts have been devoted to accelerating LLM inference. Given that inference from large models is often constrained more by memory bandwidth and communication than by arithmetic operations Leviathan et al. (2023), one innovative inference paradigm, Speculative Decoding (SD), has emerged as a new trend and shown superior performance by effectively enabling better GPU utilization. As shown in the upper part of Figure. 1, the key idea of the SD algorithm is to employ an extra small model (referred as the *draft model*) to first generate $\gamma$ draft tokens for the original large model (referred as the *target model*), and then the target model verifies these draft tokens in parallel within a single forward. Here, $\gamma$ is a fixed hyperparameter **window**

---

*This work is done when Tianyu Liu works as an intern in Tencent.
†The Corresponding Authors.

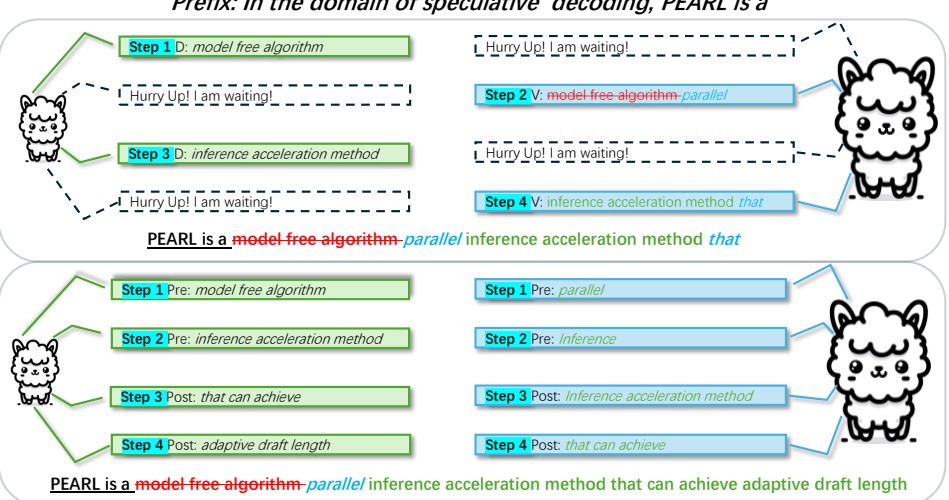

Figure 1: An overview of speculative decoding (the upper part) and our PEARL (the lower part). SD employs a draft model to provide multiple drafts and then the target model verifies the drafts in parallel. However, SD suffers from the mutual waiting problem, i.e., the target model gets stuck when the draft model is *guessing* tokens, and vice versa (the dashed dialogue box). PEARL parallels the drafting and verification process to alleviate the mutual waiting problem. Moreover, PEARL can leverage adaptive draft length to generate more tokens within the same amount of time to further mitigate the mutual waiting problem. Specifically, PEARL generates fewer draft tokens if they will be rejected (step 1 in the lower part), and more draft tokens if they can be accepted (steps 3 and 4).

**size**. **Draft length** is the number of tokens generated by the draft model in a continuous execution. Therefore, the draft length is set to $\gamma$ in SD. Following-up works effectively extend this framework by either removing the necessity of the draft model (Cai et al., 2024; Fu et al., 2024; Zhang et al., 2023) or identifying a compact draft model with high distribution alignment (Zhou et al., 2023; Zhao et al., 2024; Miao et al., 2023). Extensive experiments demonstrate that this *draft-then-verify* framework effectively enhances the concurrency of the target model, thereby significantly accelerating the inference process.

Albeit with multiple benefits of this *draft-then-verify* framework, it confronts one significant challenge that may hinder its performance and deployment—the mutual waiting problem. That is, the target model will be idle when the draft model is generating the draft tokens and the draft model will be idle when the target model is verifying the previously drafted tokens. This mutual waiting problem primarily stems from two limitations inherent in speculative decoding: **(i)** the asynchronous execution of the draft and verify phases, which directly results in the mutual waiting problem; and **(ii)** the fixed draft length, which cannot adapt to most decoding steps and thus exacerbate the mutual waiting problem.

Therefore, in this paper, we seek to answer the question: *Can we draft and verify in parallel and adaptively adjust draft length?* With this consideration, we propose a conceptually simple, flexible, and general framework to boost speculative decoding, namely **P**arallel sp**E**culative decoding with **A**daptive d**R**aft **L**ength (PEARL). Specifically, PEARL consists of two strategies *pre-verify* and *post-verify*: **(i)** *pre-verify* uses the target model to verify the first draft token during drafting phase, which allows the draft model to generate less draft tokens in difficult scenarios; **(ii)** *post-verify* uses the draft model to continue generating draft tokens during verification phase, which provides more draft tokens in simple situations. As shown in the lower part of Figure.1, PEARL effectively alleviates the mutual waiting problem with **parallelism** and **adaptive draft length** via these two strategies. We conduct extensive experiments on various text generation benchmarks, leading to a superior speed up performance up to **4.43**× and **1.50**×, compared to auto-regressive decoding and speculative decoding, respectively.

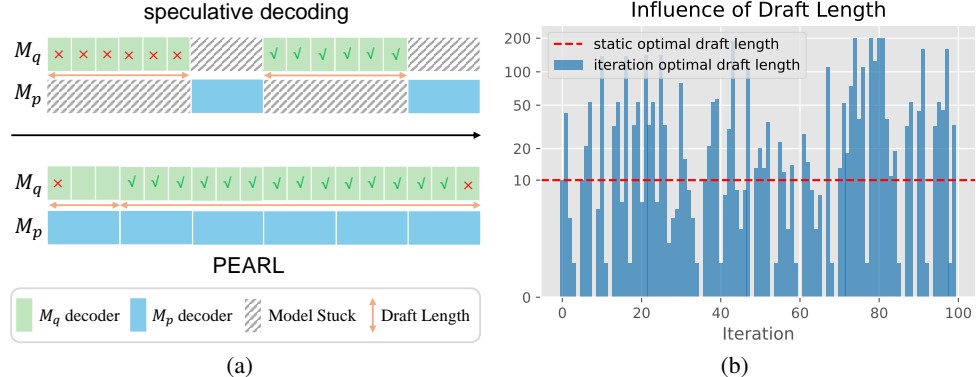

(a)                                                                    (b)

Figure 2: Motivated observations. (a) The time of both the drafting phase and verification phase is non-negligible, therefore the asynchronous execution of the draft model and the target model directly incurs the mutual waiting problem. (b) We observe that the optimal draft length changes significantly in different iterations, which exacerbates the mutual waiting problem.

## 2  BACKGROUND

**Notations.** In this paper, we use $M_q$ to denote the draft model and $M_p$ to denote the target model. $M_q(\cdot), M_p(\cdot)$ denotes the logits of the next token of a single forward of $M_q, M_p$ respectively. $\gamma$ is a hyperparameter to control the window size during speculative decoding. We denote the running speed between $M_q$ and $M_p$ as $c$, which is defined as the ratio between the time for a single forward of $M_p$ and the time for a single forward of $M_q$, i.e., $c = T(M_p(\cdot))/T(M_q(\cdot))$.

**Speculative decoding.** Given an input sequence $\mathbf{x}$ as a prefix, a speculative decoding step consists of a drafting phase and a verification phase. During the drafting phase, the draft model $M_q$ is employed to give $\gamma$ draft tokens $x_1, x_2, ..., x_\gamma$ by running $\gamma$ times model forward and sample. Here, we denote $M_q(\mathbf{x} + [x_1, ..., x_{i-1}])$ as $q_i$, then each draft token is given by $x_i \sim q_i, i = 1, ..., \gamma$. During the verification phase, the prefix $\mathbf{x}$ together with $\gamma$ draft tokens are sent to $M_p$ for verification. The target model $M_p$ inputs $\mathbf{x} + [x_1, ..., x_\gamma]$ and outputs the logits $p_1, p_2, ..., p_{\gamma+1}$. Then SD sequentially verifies $x_i$ via speculative sampling, where the acceptance rate is given by:

$$\alpha_i = \begin{cases} 1 & p_i[x_i] \geq q_i[x_i], \\ \dfrac{p_i[x_i]}{q_i[x_i]} & p_i[x_i] < q_i[x_i], \end{cases} \tag{1}$$

If SD rejects $x_i$, it will resample a token from $norm(\max(0, p_i - q_i))$, otherwise, SD accepts all the draft tokens and samples an additional token from $p_{\gamma+1}$. In this way, each SD step generates tokens with a number of at least 1 and at most $\gamma + 1$, leading to efficiency acceleration.

**Window size and draft length.** We emphasize that the window size is a hyperparameter that controls the drafting behavior. Draft length is the number of tokens generated by the draft model in a continuous execution, which is fixed and the same as the window size in SD, while draft length is adaptive and may be not equal to window size in PEARL.

## 3  METHODOLOGY

### 3.1  MOTIVATED OBSERVATION

As illustrated in Figure. 2(a), the mutual waiting problem is directly incurred by the asynchronous execution of the draft model and the target model. In our experiments, we observe that the time consumed during the drafting phase and the verification phase is usually non-negligible. Take the instance of Codellama 7B & 34B, at each decoding step, although the running speed of Codellama 7B is almost 3 times faster than Codellama 34B, the total time consumption for generating 6 draft

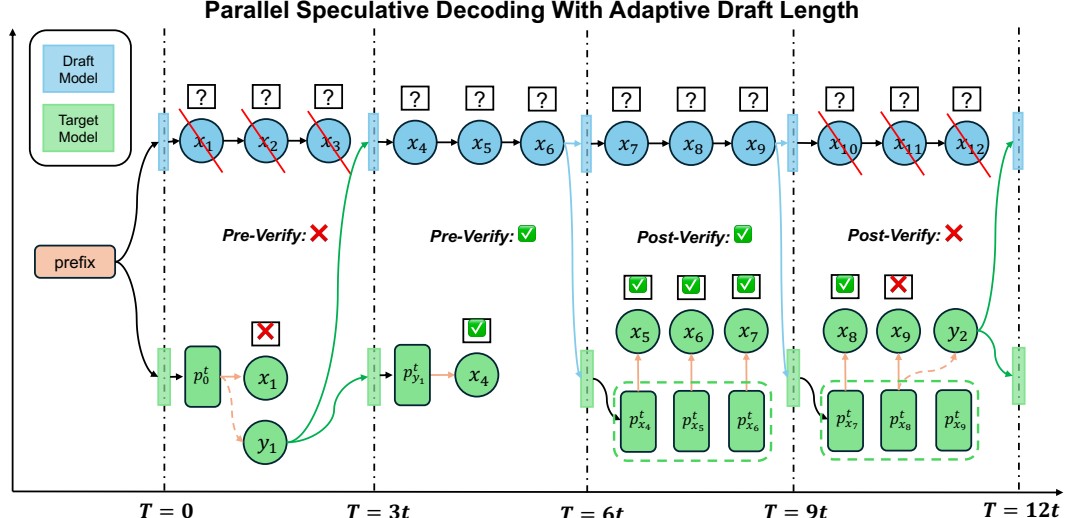

Figure 3: Illustration of our PEARL. At $T = 0$, $M_q$ generates $x_1, x_2, x_3$ and $M_p$ rejects $x_1$ with the *pre-verify* strategy. At $T = 3t$, $M_p$ accepts $x_4$ and switches to the *post-verify* strategy. At $T = 6t$, $M_p$ accepts all draft tokens $x_4, x_5, x_6$ in the last decoding step, while $M_q$ continues drafting $x_7, x_8, x_9$. At $T = 9t$, $M_p$ rejects $x_9$, drops $x_{10}, x_{11}, x_{12}$ and switches to the *pre-verify* strategy. The final output is $[y_1, x_4, x_5, x_6, x_7, x_8, y_2]$.

tokens is even 2 times than the time consumption for one verification step. Therefore, the mutual waiting problem exists **at any timestamp**, and severely affects the acceleration effectiveness of SD.

The asynchronous execution of the draft model and the target model is the direct cause of the mutual waiting problem, which is determined by two requirements of speculative decoding: (1) the drafting phase requires the input prefix to be verified; (2) the verification phase requires the draft model to complete generating draft tokens. This implies the great potential for alleviating the mutual waiting problem through parallelism: if we can remove the two requirements and parallel the drafting phase and the verification phase, a substantial acceleration can be possible.

Another limitation that aggravates the mutual waiting problem is the fixed draft length in SD, which is not appropriate for all the decoding steps. As shown in Figure 2(b), the optimal draft length changes significantly in different iterations. On the one hand, when the optimal draft length is less than the fixed draft length, the draft model will generate meaningless draft tokens that block the target model. On another hand, when the optimal draft length is more than the fixed draft length, the draft model could have generated more draft tokens that can be accepted by the target model with a single forward. However, a fixed draft length will interrupt the longer drafting phase and take an additional verification phase, which strengthens the mutual waiting problem as well. This motivates our PEARL to further alleviate the mutual waiting problem with adaptive draft length.

Together with the two motivations, we propose two simple and effective strategies, *pre-verify* and *post-verify*. The *pre-verify* removes requirement 2 and allows the target model to verify the first draft token in advance. The *post-verify* removes requirement 1 and allows the draft model to continue generating draft tokens during the verification phase. The two strategies enable parallelism and achieve adaptive draft length to effectively alleviate the mutual waiting problem.

### 3.2 PRE-VERIFY: VERIFY THE FIRST DRAFT TOKEN IN ADVANCE.

The *pre-verify* strategy aims at removing the requirement that the verification phase requires the draft model to complete generating draft tokens. Therefore, we seek to verify some draft tokens in advance during the drafting phase. We delve explicitly into the drafting stage. During the drafting phase, the draft model tries to give $\gamma$ draft tokens by running $\gamma$ times model forward. We find that the input of the draft model in $\gamma$ times forward is $\mathbf{x}, \mathbf{x} + [x_1], ..., \mathbf{x} + [x_1, x_2, ..., x_{\gamma-1}]$, respectively.

Only the origin prefix $\mathbf{x}$ can be acquired by the target model for parallel verification. Therefore, we propose to run the target model to output the logits $M_p(\mathbf{x})$ in parallel. In this way, we can verify the first token $x_1$ before the verification phase. We implement the same lossless verification method following (Leviathan et al., 2023) as illustrated in Section 2.

By applying such a *pre-verify* strategy, we can verify the first draft token before the verification phase. If the first token is rejected, all of the following draft tokens are meaningless and should be dropped. Hence we could skip the verification phase and directly conduct the next drafting phase with the prefix $\mathbf{x} + [y_1]$. If the first token is accepted, all the draft tokens will be sent to the target model in the verification phase. In Figure. 3, at the timestamp of $T = 0$, the draft model generates $x_1, x_2, x_3$ while the target model outputs $p_0^t$, rejects the first token $x_1$ and sample another token $y_1$. At the timestamp of $T = 3t$, the draft model generates $x_4, x_5, x_6$ while the target model accepts the first token $x_4$. Then $x_4, x_5, x_6$ is sent to the target model in the next verification phase.

### 3.3 POST-VERIFY: CONTINUE DRAFTING DURING VERIFICATION.

The *post-verify* strategy aims at removing the requirement that the drafting phase requires the input prefix to be verified. However, this assumption brings the limitation that the draft model should be stuck until the target model finishes verification.

Therefore, we discard this assumption and make another assumption: we directly assume that all the draft tokens can be accepted. In this way, We find that when all the $\gamma$ draft tokens are accepted, sampling a new token from $M_p(\mathbf{x} + [x_1, ..., x_\gamma])$ is not necessary, as the draft model could have generated more draft tokens that can be accepted. Hence we can use the draft model to continue drafting $x_{\gamma+1}, ..., x_{2\gamma}$ during the verification phase.

---

**Algorithm 1** Parallel Speculative Decoding with Adaptive Draft Length.

---

**Require:** the draft model $M_q$, the target model $M_p$, the input prefix $\mathbf{x}$, the max generate tokens $L$,
 the window size $\gamma$.
 Initialization: mode $\leftarrow$ "pre-verify"
 **while** $len(\mathbf{x}) < L$ **do**
  **if** mode = "pre-verify" **then**
   $\mathbf{x}$, mode $\leftarrow$ Pre-verify($M_q, M_p, \mathbf{x}, \gamma$)
  **else**
   $\mathbf{x}$, mode $\leftarrow$ Post-verify($M_q, M_p, \mathbf{x}, \gamma$)
  **end if**
 **end while**

---

If all the $\gamma$ draft tokens are accepted, we can skip the next drafting phase as we already get the draft tokens in the next drafting phase. The last logit $M_p(\mathbf{x} + [x_1, ..., x_\gamma])$ can be used to verify $x_{\gamma+1}$, which is a *"pre-verify"* process as well. In Figure. 3, at the timestamp of $T = 6t$, the target model takes in $x_4, x_5, x_6$ and outputs $p_{x_4}^t, p_{x_5}^t, p_{x_6}^t$, while the draft model continues to guess next draft tokens $x_7, x_8, x_9$. Fortunately, all the draft tokens are accepted, and we can directly conduct the next verification phase with prefix $\mathbf{x} + [y_1, x_4, x_5, x_6, x_7, x_8, x_9]$. At the timestamp of $T = 9t$, the target model takes in $x_7, x_8, x_9$ and outputs $p_{x_7}^t, p_{x_8}^t, p_{x_9}^t$, while the draft model continues to guess the next draft tokens $x_{10}, x_{11}, x_{12}$. Unfortunately, only $x_8$ is accepted, and the draft tokens $x_{10}, x_{11}, x_{12}$ will be dropped. Finally, the prefix $\mathbf{x} + [y_1, x_4, x_5, x_6, x_7, x_8, y_2]$ is input to the next drafting phase.

### 3.4 PEARL: PARALLEL SPECULATIVE DECODING WITH ADAPTIVE DRAFT LENGTH

Taking together the two strategies, our PEARL framework consists of a draft model, a target model, and two strategies to decode tokens. The two strategies are switched according to the verification results in the previous decoding step. Algorithm 1 provides a summary of our PEARL. We also provide more details in Algorithm 2. Note that pre-verify and post-verify strategies are not executed only once in the process of generating a sentence and will be repeatedly switched according to the token acceptance situation during the whole process of generating. We provide a simple step-by-step

profiling example in Appendix B for better understanding. Then we show how our PEARL achieves parallelism and adaptive draft length to alleviate the mutual waiting problem.

**Parallelism.** With the two strategies *pre-verify* and *post-verify*, At any timestamp, the draft model and the target model are running in parallel, which directly breaks the asynchronous execution of the draft model and the target model.

**Adaptive draft length.** In our PEARL, the drafting process can be seen as a segmented drafting process. If the draft model cannot generate any "right" tokens, the *pre-verify* strategy will avoid the additional drafting process. If the draft model could have generated more "right" tokens, the target model would not interrupt the drafting phase, where the draft model can generate more draft tokens with *post-verify* strategy. Therefore, PEARL can utilize the two simple yet effective strategies to implement adaptive draft length to alleviate the mutual waiting problem.

## 4 EXPERIMENTS

### 4.1 EXPERIMENTAL SETUP

**Tasks and Datasets.** We conduct experiments on various text generation tasks to evaluate the effectiveness of our PEARL, including HumanEval (code generation task) (Chen et al., 2021), GSM8K & MGSM (multilingual arithmetic reasoning task, MGSM is the multilingual translation of GSM8K) (Cobbe et al., 2021; Shi et al.), and MT-bench (multi-round dialogue task) (Zheng et al., 2024). These tasks and datasets are representative benchmarks for evaluation. More details can be found in Appendix C.1.

**Evaluation Details.** We evaluate the effectiveness of our PEARL with some state-of-the-art LLM families, including CodeLlama (Roziere et al., 2023), Deepseek-Coder (Guo et al., 2024), Llama 2 (Touvron et al., 2023) and Llama 3.1 (Dubey et al., 2024). In our experiments, the models with size less than 7B are used as the draft models and the models with size greater than 33B are used as the target models. We report the walltime speedup ratio as the metric. Additional evaluation details are provided in Appendix C.2 and C.3.

**Baseline Methods.** We implement *four training-free* inference acceleration methods as our baselines. **(i) Speculate decoding:** standalone SD methods (Leviathan et al., 2023; Chen et al., 2023) resort to a draft model to draft future tokens and then verify them in parallel. **(ii) Ouroboros:** ouroboros (Zhao et al., 2024) proposes phrase candidate pool from the verification process to generate more precise and longer drafts. **(iii) Lookahead Decoding:** look ahead decoding (Fu et al., 2024) caches the generation trajectory (n-grams) as drafts to reduce the number of total decoding steps. **(iv) Assisted generation:** assisted generation (Joao Gante, 2023) employs a heuristic approach to determine the number of draft tokens in the next iteration, based on the verification results of tokens generated by the draft model in the previous round.

Table 1: Experiment results on the code generation task. Part of the results of Lookahead Decoding and Ouroboros are taken from (Zhao et al., 2024). We **bold** the best results for each model combination. *Some results of ouroboros and lookahead decoding are reproduced in their official implementation with default parameters. Other results are reproduced in our implementation. The symbol '-' denote that the methods do not support current model configuration.[2]

| Method | CodeLlama 7&34B | CodeLlama 7&70B | Llama2 7&70B | Llama3.1 8&70B | DeepSeek 1.3&33B | DeepSeek 6.7&33B |
|---|---|---|---|---|---|---|
| Auto Regressive | 1.00× | 1.00× | 1.00× | 1.00× | 1.00× | 1.00× |
| Speculative Decoding (Leviathan et al., 2023) | 1.76× | 3.03× | 2.35× | 2.60× | 2.32× | 1.94× |
| Ouroboros (Zhao et al., 2024) | 2.14× | *3.28× | *2.10× | - | *3.25× | *2.66× |
| Lookahead Decoding (Fu et al., 2024) | 1.72× | *1.57× | *1.80× | - | *1.82× | *1.82× |
| Assisted Generation (Joao Gante, 2023) | 1.37× | 2.49× | 2.27× | 2.72× | 1.88× | 1.52× |
| **PEARL (ours)** | **2.48×** | **4.43×** | **3.29×** | **3.87×** | **3.48×** | **2.79×** |

---

[2] Their implementation requires transformers version of 4.36.2, while Llama 3.1 requires transformers $\geq$ 4.43.0

Table 2: Experiment results on the multilingual arithmetic reasoning task with Llama 2 7&70B. We **bold** the best results for each category. Results of ouroboros and lookahead decoding are reproduced in their official implementation with default parameters. Other results are reproduced in our implementation.

| Method | English (GSM8K) | Bengali | German | Spanish | French | Japanese | Russian | Swahili | Tegulu | Thai | Chinese | Avg. |
|---|---|---|---|---|---|---|---|---|---|---|---|---|
| Auto Regressive | 1.00× | 1.00× | 1.00× | 1.00× | 1.00× | 1.00× | 1.00× | 1.00× | 1.00× | 1.00× | 1.00× | 1.00× |
| Speculative Decoding | 2.48× | 2.69× | 2.77× | 2.64× | 2.71× | 2.71× | 2.72× | 2.81× | 2.65× | 2.71× | 2.78× | 2.70× |
| Ouroboros | 1.60× | 1.75× | 1.88× | 1.69× | 1.80× | 1.95× | 1.65× | 1.68× | 2.45× | 1.92× | 1.81× | 1.84× |
| Lookahead Decoding | 1.23× | 1.34× | 1.51× | 1.50× | 1.48× | 1.29× | 1.43× | 1.60× | 1.28× | 1.23× | 1.48× | 1.39× |
| Assisted Generation | 1.96× | 1.69× | 1.75× | 1.70× | 1.67× | 2.02× | 1.68× | 1.58× | 3.07× | 2.17× | 1.97× | 1.93× |
| **PEARL (ours)** | **3.82×** | **3.94×** | **4.00×** | **3.81×** | **3.76×** | **3.94×** | **3.85×** | **4.18×** | **4.10×** | **3.93×** | **4.06×** | **3.95×** |

Table 3: Experiment results on the multi-round dialogue task with Llama 2 7&70B. We **bold** the best results for each category. Results of ouroboros and lookahead decoding are reproduced in their official implementation with default parameters. Other results are reproduced in our implementation.

| Method | Writing | Roleplay | Reasoning | Math | Coding | Extraction | Stem | Humanities | Avg. |
|---|---|---|---|---|---|---|---|---|---|
| Auto Regressive | 1.00× | 1.00× | 1.00× | 1.00× | 1.00× | 1.00× | 1.00× | 1.00× | 1.00× |
| Speculative Decoding | 1.70× | 1.73× | 1.96× | 2.00× | 1.93× | 2.14× | 1.87× | 1.81× | 1.89× |
| Ouroboros | 1.42× | 1.35× | 1.40× | 1.61× | 1.35× | 1.67× | 1.44× | 1.36× | 1.45× |
| Lookahead Decoding | 1.31× | 1.24× | 1.50× | 1.51× | 1.38× | 1.40× | 1.29× | 1.27× | 1.36× |
| Assisted Generation | 1.41× | 1.40× | 1.39× | 1.64× | 1.74× | 1.92× | 1.57× | 1.47× | 1.55× |
| **PEARL (ours)** | **2.40×** | **2.45×** | **2.85×** | **2.79×** | **2.67×** | **2.92×** | **2.58×** | **2.50×** | **2.64×** |

## 4.2 MAIN RESULTS.

We conduct extensive experiments on the aforementioned benchmarks. As shown in Table 1, PEARL significantly outperforms vanilla speculative decoding, Ouroboros, Lookahead decoding and assisted generation in all backbone model configurations on the HumanEval dataset, which encompass different scales of model configurations including 1.3&33B, 6.7&33B (7&34B) and 7&70B. Specifically, PEARL can achieve up to $4.43\times$ and $1.50\times$ speed up compared with vanilla auto-regressive methods and vanilla speculative decoding, respectively. These results indicate the universal existence of the mutual waiting problem, and demonstrate that PEARL effectively addresses the mutual waiting problem, thereby achieving significant inference acceleration results compared to methods based on the traditional draft-then-verify framework. Moreover, as shown in Table 2, 3, PEARL can also achieve significant inference acceleration on 12 multilingual arithmetic tasks and 8 multi-round dialogue tasks, whereas PEARL can achieve $1.36 \sim 1.55\times$ speedup compared with vanilla speculative decoding. These results demonstrate the superior potential of the parallel speculative decoding framework to exploit the computation resources more adequately. We provide more evaluation results on MGSM and MT-bench with more advanced Llama 3.1 8&70B in Appendix D.

## 4.3 ABLATION STUDIES

To provide more insights into the two proposed strategies, we conduct the ablation study. We denote PEARL without *pre-verify* as PEARL *w/o pre-verify* and PEARL without *post-verify* as PEARL *w/o post-verify* and present the main results of ablation studies.

As shown in Table 4, the absence of any strategy of PEARL results in a performance degradation of the entire framework. The absence of the *post-verify* strategy exhibits a more pronounced impact on the performance of PEARL than the *pre-verify* strategy. We explain the reason for this phenomenon as follows. Intuitively, the *pre-verify* strategy makes more contributions when the acceptance rate is relatively low. The *pre-verify* strategy can save a target model forward when the first draft token is rejected by the target model. Denote the acceptance rate as $\alpha$, and the *pre-verify* strategy will take effect with probability $1 - \alpha$. Therefore, better alignment between the draft model and the target model will make *pre-verify* strategy less effective. However, the *post-verify* strategy makes more contributions when the two models are aligned, i.e., there are more situations in which all draft tokens are accepted by the target model. Therefore, the two strategies are complementary and accelerate inference together.

Table 4: Ablation results of PEARL on HumanEval and GSM8K datasets.

| | HumanEval | | | GSM8K |
|---|---|---|---|---|
| **Methods** | **CodeLlama 7B&34B** | **CodeLlama 7B&70B** | **DeepSeek 1.3B&33B** | **Llama 2 7B&70B** |
| PEARL *w/o pre-verify* | 2.21× (↓ 0.14) | 3.53× (↓ 0.26) | 3.19× (↓ 0.29) | 2.51× (↓ 0.26) |
| PEARL *w/o post-verify* | 1.64× (↓ 0.71) | 2.57× (↓ 1.22) | 2.37× (↓ 1.11) | 2.15× (↓ 0.72) |
| PEARL | **2.35×** | **3.79×** | **3.48×** | **2.87×** |

In our experiments, all the model combinations show great alignment, which leads to the superiority of the *post-verify* strategy. As the language models evolve and more speculative decoding methods, the alignment between the draft model and the target model will be better, which further highlights the importance of the *post-verify* strategy. Meanwhile, we can further improve the *pre-verify* strategy by pre-verifying multiple draft tokens (similar to the cache pool in Ouroboros and Lookahead Decoding) for more acceleration. We leave these as future works.

## 4.4 CASE STUDIES

In this subsection, we present two important case studies to discuss the sensitive analysis of $\gamma$ and the mean accepted tokens of PEARL. We provide more experiment results in Appendix D.

### 4.4.1 SENSITIVE ANALYSIS OF THE WINDOW SIZE $\gamma$

Intuitively, the optimal value of the window size $\gamma$ should be the speed ratio $c$ between the draft model and the target model, where the draft model and the target model can achieve best parallelism and fully alleviate the mutual waiting problem. However, it is often the case that $c$ may not be an integer. Therefore, we propose to select the round integer of $c$ as the window size $\gamma$. We conduct some case studies to see the effect of different $\gamma$ values in different model configurations and different tasks. As shown in Table 5 (we mark the round integer below each model configuration), directly choosing the round integer of $c$ as the window size can achieve the maximal inference acceleration, which is robust to the model configurations. Meanwhile, as shown in Table 6 (the round integer of $c$ of Llama 2 7&70B is 5), setting the window size as 5 can achieve the maximal inference acceleration as well, which is robust to the tasks. These results suggest great convenience of PEARL framework which alleviates the burden of tuning $\gamma$ for different model configurations and different tasks.

Table 5: Optimal $\gamma$ of different model combinations on HumanEval. (unit: tok/sec)

| $\gamma$ | CodeLlama 7&34 (c=3) | CodeLlama 7&70 (c=5) | DeepSeek 6.7&33 (c=3) |
|---|---|---|---|
| 2 | 33.25 | 16.28 | 30.82 |
| 3 | **46.06** | 23.14 | **48.46** |
| 4 | 44.12 | 29.65 | 47.22 |
| 5 | 44.93 | **40.72** | 46.91 |
| 6 | 41.83 | 35.39 | 44.36 |

Table 6: Optimal $\gamma$ for different tasks of Llama 2 7B&70B. (c=5)

| $\gamma$ | **HumanEval** | **GSM8K** | **MT-Bench** |
|---|---|---|---|
| 3 | 20.39 | 18.23 | 17.67 |
| 4 | 24.58 | 21.81 | 20.69 |
| **5** | **30.34** | **26.47** | **24.25** |
| 6 | 28.02 | 24.59 | 22.71 |
| 7 | 28.09 | 24.23 | 22.54 |

### 4.4.2 MEAN ACCEPTED TOKENS

In Section 3.4, we claim that PEARL can achieve adaptive draft length for acceleration. To further illustrate the real mean accepted tokens in PEARL under real-world complex conditions, we conduct experiments on the HumanEval, GSM8K, and MT-Bench datasets. As shown in Table 7, we still empirically observe that that PEARL obtains more accepted tokens compared to vanilla SD methods, which further demonstrates the effectiveness of the PEARL framework. Specifically, PEARL achieves the max number of mean accepted tokens to **39.9**, which significantly outperforms vanilla SD methods by a large margin. Note that the mean accepted tokens (**MAT**) and the speed ratio c between the draft model and the target model both influence the final speed-up results. For example, in the case of Deepseek 6.7&33B, the draft model runs approximately three times faster than the target model. Even if the MAT approaches infinity, where all tokens are generated by the 6.7B model, the theoretical maximum speed-up would be capped at 3×. Consequently, with a MAT of 39.9, PEARL

achieves a 2.75× speed-up, which is close to this theoretical optimum. These results demonstrate that our PEARL can fully exploit the inference ability of the draft model for further acceleration.

Table 7: Comparison of mean accepted tokens of vanilla SD methods and PEARL.

| Methods | CodeLlama 7B&34B | CodeLlama 7B&70B | DeepSeek 1.3B&33B | DeepSeek 6.7B&33B |
|---|---|---|---|---|
| Speculative Decoding | 5.27 | 8.32 | 7.23 | 5.69 |
| **Ours** | **27.95** | **26.53** | **29.65** | **39.90** |

## 4.5 MORE DISCUSSIONS ON PEARL

**Clarification of the application scenarios.** First, we would like to clarify the application scenarios of our PEARL. The key idea of speculative decoding is to exploit the computation redundancy for acceleration Leviathan et al. (2023). Based on this idea, we observe the mutual waiting problem, which hinders speculative decoding to fully utilize the redundant computational resources. Therefore, the main application scenarios of PEARL focus on the scenarios with adequate computational resources, where speculative decoding cannot sufficiently use these resources.

**PEARL in resource-adequate scenarios.** In such scenarios, the draft model and the target model can be deployed separately. Simultaneously running the draft model and the target model would not bring additional latency in either both drafting or verification stages. Our main experiments along with all baseline methods are conducted under these scenarios, where we deploy the draft model and the target model on different devices. Besides, it is feasible to integrate tensor parallelism (TP) with PEARL in these situations. We provide the solution in Appendix F.

**PEARL in resource-constrained scenarios.** However, we acknowledge that in many situations, the GPU resources are limited, and the draft model and the target model are deployed on the same devices. We refer to this as a "co-locate" setting or resource competitions (RC). The key problem lies in the nature of GPU hardware design—two running processes on the same GPU will compete for GPU resources, which may lead to slowdowns. We provide a solution to address this issue.

Generally, in real-world LLM applications, the large-scale target model is usually placed with more than 1 GPU to handle more requests and long context inference, while the small-scale draft model only needs 1 GPU for inference. In this case, we can apply pipeline parallelism (PP) to serve the target model with multiple GPUs. Inspired by this observation, we propose an improved version of PEARL to effectively utilize GPU computation resources with PP without resource competitions. The key idea is to transfer the computation of the draft model to another GPU when the target model is running on a specific GPU. Specifically, we transfer the first $\lceil \frac{\gamma}{2} \rceil$ draft token generation to the last device, while the last $\lfloor \frac{\gamma}{2} \rfloor$ draft tokens are generated with the first device. As the computation of the target model is conducted sequentially with multiple GPUs, this could effectively utilize the GPU resources to avoid RC. We conduct some experiments in Table 8 and find that this strategy allows PEARL to retain 89% ∼ 99% of its original performance, demonstrating the effectiveness of our PEARL in such conditions. We provide detailed implementation and additional experiment results of this strategy in Appendix G.

Table 8: Comparisons of Llama models for PEARL on MT-bench with and without RC scenario.

| Models | Writing | Roleplay | Reasoning | Math | Coding | Extraction | Stem | Humanities | Avg. |
|---|---|---|---|---|---|---|---|---|---|
| Llama 2 7b&70b | 22.10 | 22.47 | 26.16 | 25.74 | 24.63 | 26.11 | 23.84 | 23.09 | 24.28 |
| Llama 2 7b&70b (RC) | 19.88 | 20.24 | 25.06 | 24.47 | 23.47 | 25.79 | 21.55 | 22.03 | 22.83 |
| performance retain | 89.95% | 90.08% | 95.80% | 95.07% | 95.29% | 98.77% | 90.39% | 95.41% | 94.03% |
| Llama 3.1 8b&70b | 31.23 | 30.08 | 35.09 | 36.59 | 31.95 | 34.60 | 30.06 | 27.51 | 32.14 |
| Llama 3.1 8b&70b (RC) | 29.65 | 27.54 | 35.01 | 36.31 | 29.85 | 33.99 | 26.77 | 26.10 | 30.78 |
| performance retain | 94.94% | 91.56% | 99.77% | 99.23% | 93.43% | 98.24% | 89.06% | 94.87% | 95.77% |

## 5 RELATED WORK

**Transformer inference acceleration.** Inference acceleration is a field that has been extensively studied over a long period of time (Liu et al., 2024). There exists extensive works for transformer

inference acceleration with the rising of LLMs (Xia et al., 2024; Lv et al., 2024; Chen et al., 2024). This includes efforts of model compression (Zhu et al., 2023), efficient architecture design (Chitty-Venkata & Somani, 2022), and hardware optimization and implementation (Dao et al., 2022). Model compression methods such as quantization (Choi et al., 2018), knowledge distillation (Hinton et al., 2015), and structure pruning (Han et al., 2015) aim at reducing the number of computational operations. Efficient architecture design is proposed to develop lightweight transformer architectures. Hardware optimization and implementation is proposed for efficient execution to fully exploit the hardware devices. These methods have achieved great success, while they are orthogonal to speculative decoding algorithms, which can be integrated for further speedup.

**Draft-then-verify framework.** While SD exhibits great acceleration effectiveness and lossless generalization quality, it remains a challenge to find a compact draft model with high distribution alignment. Some works focus on removing the necessity of the draft model. Self-speculative decoding (Zhang et al., 2023) proposes to skip some intermediate layers of the target model for drafting. Medusa (Cai et al., 2024) adds extra decoding heads at the top of the target model to generate drafts. Lookahead decoding(Fu et al., 2024) caches the generation trajectory (n-grams) as the drafts. Eagle (Li et al., 2024) employs an additional transformer decoder layer to generate drafts at the feature level. Glide (Du et al., 2024) reuses the kv cache from the target model to decode more accurate draft tokens. DistillSpec (Zhou et al., 2023) utilizes distillation method to identify a compact draft model. Ouroboros (Zhao et al., 2024) combines the standard SD and lookahead decoding to generate more precise and longer drafts. Besides these works, SpecInfer (Miao et al., 2023) proposes tree attention, which is widely used to verify more drafts and increase the acceptance rate. However, all of them do not address the parallelism issue. From this perspective, our PEARL is orthogonal to these methods and can be integrated with these methods, which is left as a future work.

# 6  CONCLUSION AND FUTURE WORK

**Limitations and broader impact.** As our PEARL is a parallel acceleration framework, it remains a challenge to schedule the GPU resources to avoid resource competitions, which may potentially increase power consumption. We affirm our commitment to contributing positively to society, avoiding harm, and upholding honesty and trustworthiness. We appropriately cite the previous methods and datasets we use, and ensure that all data involved is fully public, with no private data being utilized. Furthermore, we are committed to correctly maintaining the inference acceleration techniques we have developed, without incurring any form of discrimination.

**Conclusion.** In this paper, we propose a novel inference acceleration framework, called PEARL, which significantly improves LLM inference efficiency. PEARL consists of two simple and effective strategies, i.e., *pre-verify* and *post-verify*, which effectively alleviates the mutual waiting problem with parallelism and adaptive draft length. Extensive experiments demonstrate that our proposed PEARL outperforms existing state-of-the-art methods on various text generation benchmarks.

**Future work.** For future research, we aim to integrate PEARL with existing accelerated inference methods to explore more efficient and resource-friendly acceleration approaches for LLM inference. Hopefully, PEARL will facilitate the future development of LLM inference acceleration.

## ACKNOWLEDGEMENTS

The authors would like to thank all the anonymous reviewers for their insightful comments. This paper is supported by Shanghai Artificial Intelligence Laboratory.

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

## A  ALGORITHM OF PEARL

Here, we give the whole algorithm of our PEARL in detail in Algorithm. 2.

---

**Algorithm 2** Parallel Speculative Decoding with Adaptive Draft Length.

---

**Input:** the draft model $M_q$, the target model $M_p$, the input prefix $\mathbf{x}$, the max generate tokens $L$, the window size $\gamma$.

    ▷ The *pre-verify* strategy is used first.

1: Initialization: mode ← "pre-verify"
2: **while** $len(\mathbf{x}) < L$ **do**
3:   **if** mode = "pre-verify" **then**
4:     ▷ Pre-verify strategy
5:     **for** $i = 1$ to $\gamma$ **do**
6:       $q_i \leftarrow M_q(\mathbf{x} + [x_1, ..., x_{i-1}])$
7:       $x_i \sim q_i$
8:     **end for**
9:     ▷ running the target model in parallel to verify the first draft token in advance.
10:     $p \leftarrow M_p(\mathbf{x})$
11:     **if** $r \sim U(0,1) \leq \frac{p[x_1]}{q_1[x_1]}$ **then**
12:       ✓ accept the first token
13:       $\mathbf{x} \leftarrow \mathbf{x} + [x_1, ..., x_\gamma]$
14:       mode ← "post-verify"
15:     **else**
16:       × reject the first token
17:       $y \sim norm(max(0, p - q_1))$
18:       $\mathbf{x} \leftarrow \mathbf{x} + [y]$
19:       mode ← "pre-verify"
20:     **end if**
21:   **else**
22:     ▷ Post-verify strategy
23:     $\mathbf{x}, [x_1, x_2, ..., x_\gamma] \leftarrow \mathbf{x}$       ▷ split the prefix to get the last $\gamma$ draft tokens
24:     **for** $i = \gamma + 1$ to $2\gamma$ **do**
25:       ▷ running the draft model in parallel to continue drafting.
26:       $q_i \leftarrow M_q(\mathbf{x} + [x_1, ..., x_{i-1}])$
27:       $x_i \sim q_i$
28:     **end for**
29:     $p_1, p_2, ..., p_\gamma \leftarrow M_p(\mathbf{x} + [x_1]), M_p(\mathbf{x} + [x_1, x_2]), ..., M_p(\mathbf{x} + [x_1, ..., x_\gamma])$
30:     retrival $q_1, q_2, ..., q_\gamma$ from the cache
31:     $r_1 \sim U(0,1), ..., r_\gamma \sim U(0,1)$
32:     $n \leftarrow \min(\{i - 1 | 1 \leq i \leq \gamma, r_i > \frac{p_i[x_i]}{q_i[x_i]}\} \cup \{\gamma\})$
33:     **if** $n = \gamma$ **then**
34:       ✓ accept all draft tokens
35:       $\mathbf{x} \leftarrow \mathbf{x} + [x_1, ..., x_{2\gamma}]$
36:       mode ← "post-verify"
37:     **else**
38:       × reject someone
39:       $y \sim norm(max(0, p_{n+1} - q_{n+1}))$
40:       $\mathbf{x} \leftarrow \mathbf{x} + [x_1, ..., x_n, y]$
41:       mode ← "pre-verify"
42:     **end if**
43:   **end if**
44: **end while**

---

## B  A SIMPLE STEP-BY-STEP PROFILING EXAMPLE

We provide a simple step-by-step profiling of PEARL with a real data prompt "x+y = 4z, x*y = 4z^2, express x-y in z" in Table 9.

Table 9: Simple step-by-step profiling of PEARL with prompt "x+y = 4z, x*y = 4z^2, express x-y in z". We only report the first 7 steps for simplicity. The prompt is selected from MT-bench, and we use Llama 2 7&70b as our base model pair.

| steps | input prefix | current mode | draft model output | target model output | judging reason | output prefix |
|---|---|---|---|---|---|---|
| 0 | x+y = 4z, x*y = 4z^2, express x-y in z | pre-verify | Great, I'm | I | great is not I, hence great is rejected, turn to pre-verify | I |
| 1 | I | pre-verify | 'm glad you | ' | ' is accepted, turn to post-verify | I' |
| 2 | I'm glad you | post-verify | are interested in exploring | m | m is accepted, but glad is rejected, turn to pre-verify | I'm happy |
| 3 | I'm happy | pre-verify | to help you with this | to | to is accepted, turn to post-verify | I'm happy to |
| 4 | I'm happy to help you with this | post-verify | equation! However, I | help you with | the first 3 tokens are accepted, but this is rejected, turn to pre-verify | I'm happy to help you with your |
| 5 | I'm happy to help you with your | pre-verify | question! However, I | question | question is accepted, turn to post-verify | I'm happy to help you with your question |
| 6 | I'm happy to help you with your question! However, I | post-verify | notice that the equation you | ! However, I notice | all previous draft tokens are accepted, keep post-verify | I'm happy to help you with your question! However, I notice |
| 7 | I'm happy to help you with your question! However, I notice that the equation you | post-verify | provided is not correct. | that the | equation is rejected. turn to pre-verify | I'm happy to help you with your question! However, I notice that the equations |

we give some explanations about the whole process.

1) At step 0, the prompt "x+y = 4z, x*y = 4z^2, express x-y in z" is input to the draft model and the target model simultaneously with strategy pre-verify. Within the left of this explanation, we omit this original prompt for simplicity. The draft model outputs [Great, I'm], while the target model outputs [I]. Then, we will use [I] to verify the first token [Great] in the draft tokens. As it

is not the same, we reject the draft tokens, save a verification stage of the other draft tokens for acceleration, and the output prefix is [I]. As there exists a rejected token, the next strategy is still pre-verify.

2) At step 1, the prefix [I] is input, and the draft model outputs ['m glad you] and the target model outputs [']. This time, ['] is accepted by the target model, and the next strategy is tuned to post-verify.

3) At step 2, the prefix together with the other draft tokens [I'm glad you] is input. The draft model outputs [are interested in exploring] while the target model outputs [m], i.e., the first draft token [m] is accepted, but the second draft token [glad] is rejected. The target model additionally appends [happy] to the prefix. As there exists a rejected token, the next strategy is pre-verify.

4) At step 3, the prefix [I'm happy] is input, and the draft model outputs [to help you with this] and the target model outputs [to]. This time, [to] is accepted by the target model, and the next strategy is tuned to post-verify.

5) At step 4, the prefix together with the other draft tokens [I'm happy to help you with this] is input. The draft model outputs [equation! However, I] while the target model outputs [help you with], i.e., the first three draft tokens [help you with] is accepted, but the fourth draft token [you] is rejected. The target model additionally appends [your] to the prefix. As there exists a rejected token, the next strategy is pre-verify.

6) At step 5, the prefix [I'm happy to help you with your] is input, and the draft model outputs [question! However, I] and the target model outputs [question]. This time, [question] is accepted by the target model, and the next strategy is tuned to post-verify.

7) At step 6, the prefix together with the other draft tokens [I'm happy to help you with your question! However, I] is input. The draft model outputs [notice that the equation you] while the target model outputs [! However, I notice]. All the draft tokens are accepted, and the next strategy is still post-verify. Therefore, we save the times of the draft model forward for acceleration.

8) At step 7, the prefix together with the other draft tokens [I'm happy to help you with your question! However, I notice that the equation you] is input. The draft model outputs [provided is not correct] while the target model outputs [that the], i.e., the first two draft tokens [that the] are accepted, but the third draft token [equation] is rejected. The target model additionally appends [equations] to the prefix. As there exists a rejected token, the next strategy is pre-verify.

## C  EVALUATION DETAILS

### C.1  DATASET CONFIGURATIONS

In our experiments, we evaluate the effectiveness of our PEARL on 4 categories of text generation tasks, including code generation, arithmetic reasoning, multilingual inference, and multi-round dialogue. For the code generation task, we employ HumanEval (Chen et al., 2021), a famous code generation benchmark which is composed of 164 entries. For arithmetic reasoning and multilingual inference, we employ GSM8K and MGSM (Cobbe et al., 2021; Shi et al.) as the evaluation benchmark. As the GSM8K is the English version of MGSM, we report their results in the same table. For GSM8K, we sample the first 100 entries for evaluation. For the other 10 categories in MGSM, we select 10 entries for each language. For multi-round dialogue, we employ MT-bench(Zheng et al., 2024) as the benchmark. The maximum generation lengths of these tasks are respectively set to 1024, 256, 256, and 256.

### C.2  MODEL CONFIGURATIONS

We select some representative models for evaluation, including Llama 2 Touvron et al. (2023), Codellama Roziere et al. (2023), and Deepseek-Coder Guo et al. (2024). We summarize the model configuration in Table 10. In our experiments, all models are loaded in the precision of bfloat-16. Our PEARL does not introduce any additional training, and directly uses these models to evaluate our algorithm. The running speed is measured on the code generation tasks.

Table 10: Detailed model configurations.

| Models | Layers | dim | FFN dim | speed (tok/s) |
|---|---|---|---|---|
| Codellama-7B | 32 | 4096 | 11008 | 49.34 |
| Codellama-34B | 48 | 8192 | 22016 | 18.58 |
| Codellama-70B | 80 | 8192 | 28672 | 9.20 |
| Deepseek-1.3B | 24 | 2048 | 5504 | 63.20 |
| Deepseek-6.7B | 32 | 4096 | 11008 | 50.05 |
| Deepseek-33B | 62 | 7168 | 19200 | 17.37 |
| Llama-2-7B | 32 | 4096 | 11008 | 49.94 |
| Llama-2-70B | 80 | 8192 | 28672 | 9.22 |
| Llama-3.1-8B | 32 | 4096 | 14336 | 44.37 |
| Llama-3.1-70B | 80 | 8192 | 28672 | 9.00 |

## C.3 EVALUATION DETAILS

All of our experiments including latency measurement, ablation studies, and case studies are conducted on NVIDIA A100-SXM4-80G GPUs. For models with sizes of 1.3B and 7B, we put them on a single A100, while 34B models are deployed on 2 A100, and 70B models are deployed on 3 A100. For inference, we use batch size 1, which is commonly used in other speculative decoding works. For the compared baselines, including Lookahead decoding and Ouroboros, we reproduce the results of them on the code generation tasks with the default parameters as described in their paper or code. When evaluating these methods, the model configuration and GPU usage are the same as our PEARL.

Table 11: The number of model runs of the draft model and the target model with different model configurations on HumanEval

| | Draft Model (SD) | Target Model (SD) | Draft Model (PEARL) | Target Model (PEARL) |
|---|---|---|---|---|
| Deepseek 1.3B&33B | 140500 | 35125 | 181864 (1.29×) | 45466 (1.29×) |
| Deepseek 6.7B&33B | 128973 | 42991 | 174855 (1.35×) | 58285 (1.36×) |
| Codellama 7B&34B | 132054 | 44018 | 181020 (1.37×) | 60340 (1.37×) |
| Codellama 7B&70B | 151960 | 30392 | 198370 (1.30×) | 39674 (1.30×) |
| Llama2 7B&70B | 175460 | 35092 | 248720 (1.41×) | 49744 (1.42×) |

As our PEARL is a parallel inference acceleration framework, we implement the parallel algorithm in accelerate, which can be further optimized with other parallel techniques. We leave this as a potential future work to acquire more acceleration.

# D MORE EXPERIMENT RESULTS

## D.1 EVALUATION RESULTS OF LLAMA 3.1 ON MT-BENCH AND MGSM

As illustrated in Section 4.2, we provide more evaluation results of PEARL in Table 12 and 13 with both Llama 2 7&70B and Llama 3.1 8&70B. Notably, Llama 3.1 is a more advanced LLM series which requires the transformers version to be greater than 4.43.0. Therefore, we cannot reproduce the results of baseline Ouroboros and Lookahead Decoding.

Table 12: Multi-language experiment results using Llama 3.1 8B&70B on GSM8K and MGSM (Cobbe et al., 2021; Shi et al.). We **bold** the best results for each language.

| Method | English (GSM8K) | Bengali | German | Spanish | French | Japanese | Russian | Swahili | Tegulu | Thai | Chinese | Avg. |
|---|---|---|---|---|---|---|---|---|---|---|---|---|
| Auto Regressive | 1.00× | 1.00× | 1.00× | 1.00× | 1.00× | 1.00× | 1.00× | 1.00× | 1.00× | 1.00× | 1.00× | 1.00× |
| Speculative Decoding | 2.48× | 2.69× | 2.77× | 2.64× | 2.71× | 2.71× | 2.72× | 2.81× | 2.65× | 2.71× | 2.78× | 2.70× |
| **Ours** | **3.82×** | **3.94×** | **4.00×** | **3.81×** | **3.76×** | **3.94×** | **3.85×** | **4.18×** | **4.10×** | **3.93×** | **4.06×** | **3.95×** |

Table 13: Experiment results using Llama2 7B&70B and Llama 3.1 8B&70B on MT-bench (Zheng et al., 2024). We **bold** the best results for each subtask.

| Model Configuration | Method | Writing | Roleplay | Reasoning | Math | Coding | Extraction | Stem | Humanities | Avg. |
|---|---|---|---|---|---|---|---|---|---|---|
| | Auto Regressive | 1.00× | 1.00× | 1.00× | 1.00× | 1.00× | 1.00× | 1.00× | 1.00× | 1.00× |
| Llama 2 7B&70B | Speculative Decoding | 1.70× | 1.73× | 1.96× | 2.00× | 1.93× | 2.14× | 1.87× | 1.81× | 1.89× |
| | **Ours** | **2.40×** | **2.45×** | **2.85×** | **2.79×** | **2.67×** | **2.92×** | **2.58×** | **2.50×** | **2.64×** |
| | Auto Regressive | 1.00× | 1.00× | 1.00× | 1.00× | 1.00× | 1.00× | 1.00× | 1.00× | 1.00× |
| Llama 3.1 8B&70B | Speculative Decoding | 2.29× | 2.24× | 2.66× | 2.81× | 2.35× | 2.64× | 2.22× | 2.12× | 2.42× |
| | **Ours** | **3.49×** | **3.35×** | **3.92×** | **4.06×** | **3.55×** | **3.95×** | **3.34×** | **3.05×** | **3.59×** |

## D.2 OPTIMAL $\gamma$ OF SPECULATIVE DECODING

In recent speculative decoding papers, the compared results of vanilla speculative decoding are commonly based on a fixed window size $\gamma = 5$. We find that vanilla speculative decoding can achieve better results with appropriate $\gamma$. Therefore, all the results of speculative decoding in our paper are based on their optimal $\gamma$. We present some searching results of $\gamma$ for speculative decoding in Table 14.

Table 14: Optimal $\gamma$ values of speculative decoding for each model pair. (Unit: tokens / second)

| codellama 7&34B | codellama 7&70B | deepseek 1.3&33B | deepseek 6.7&33B |
|---|---|---|---|
| 30.95 ($\gamma = 4$) | 25.92 ($\gamma = 8$) | 37.22 ($\gamma = 6$) | 30.92 ($\gamma = 4$) |
| 31.85 ($\gamma = 5$) | 26.02 ($\gamma = 9$) | 38.53 ($\gamma = 7$) | 31.74 ($\gamma = 5$) |
| **33.57** ($\gamma = 6$) | **27.60** ($\gamma = 10$) | **39.52** ($\gamma = 8$) | **33.77** ($\gamma = 6$) |
| 33.52 ($\gamma = 7$) | 27.23 ($\gamma = 11$) | 39.38 ($\gamma = 9$) | 33.55 ($\gamma = 7$) |
| 32.79 ($\gamma = 8$) | 26.65 ($\gamma = 12$) | 38.69 ($\gamma = 10$) | 32.97 ($\gamma = 8$) |

## D.3 TIME CONSUMPTION OF EACH COMPONENT IN ONE PEARL STEP

To investigate the influence and potential additional latency of the parallel inference, we measure the time cost of each component in one PEARL step with different sizes of model pairs in Table 15. From these results, we can find that the communication cost is negligible. The draft time and the target time are very close, indicating that PEARL effectively addresses the mutual waiting problem.

Table 15: The time cost of each component in one PEARL step. The experiments are conducted on HumanEval.

| | llama 2 7&70B | codellama 7&34B | deepseek 1.3&33B |
|---|---|---|---|
| **communication** | 0.2 ms | 0.3 ms | 0.2 ms |
| **verify** | 1.7 ms | 1.6 ms | 1.7 ms |
| **draft** | 105.1 ms | 68.9 ms | 65.1 ms |
| **target** | 108.0 ms | 71.1 ms | 66.1 ms |

## E FORWARD TIMES COMPARISON OF PEARL AND SD METHODS

Considering that PEARL is a parallel framework, both the draft model and the target model are running simultaneously at all times. Therefore, we measure the number of model runs for PEARL compared to the traditional SD method to provide a more comprehensive perspective in Table 11. The results show that our PEARL exhibits relatively more forward times of both the draft model and the target model compared to traditional SD. As our PEARL is a parallel inference framework, which executes the draft model and the target model in parallel at any timestamp, it naturally increases the

forward times of the target model and leads to more power consumption. However, the additional inference time occurs at another process, which will not affect the multi-user throughput.

## F  INTEGRATING TP WITH PEARL

In resource-adequate scenarios, it is possible to integrate tensor parallelism (TP) with PEARL. The key to integrating TP is to deploy the draft model and the target model on separate devices. The most direct way is to deploy the small-scale draft model on 1 GPU and the large-scale target model on the rest GPUs. Take the example of 8 GPUs, we can place the draft model on GPU 0, while the target model is set on GPUs 1-7. In this way, the draft model and the target model can conduct parallel inference and achieve the best inference speedup. Meanwhile, it is possible to deploy the draft model on CPU / edge computing. The separation idea is similar to PD dis-aggregation (Zhong et al., 2024), which dis-aggregates the prefilling and decoding process on different devices to fully exploit the computation resources. **PEARL shares the same idea to disaggregate the decoding process of the draft model and the target model on different devices.**

However, we notice that in the modern TP framework, layer width should be divisible by TP size (which would not work with TP=7 as shown in the example. We propose to use padding techniques to address the division issue directly. For example, given a weight matrix $W \in \mathcal{R}^{4096 \times 4096}$ and TP=7, we can append an extra zero matrix $W_{pad} \in \mathcal{R}^{4096 \times 6}$ to $W$, and form a padded weight matrix $\hat{W} \in \mathcal{R}^{4096 \times 4102}$. In this way, the dimension issue can be effectively addressed. For other $W$ and TP size pairs, we can get the padded matrix similarly. We provide some explanation to show this padding technique is lossless to the performance.

1. For MLP layers, padding a zero matrix $W_{pad} \in \mathcal{R}^{d \times r}$ to the weight matrix does not affect the final results. Directly remove the final $r$ columns of the output can get the original output.

2. For attention layers, padding a zero matrix $W_{pad} \in \mathcal{R}^{d \times r}$ to the weight matrix is equivalent to padding a zero matrix to $Q, K$, where $Q_{pad} = Q[W^Q; W_{pad}^Q], K_{pad} = K[W^K; W_{pad}^K]$. The attention weight is computed as $Q_{pad} K_{pad}^T = Q[W^Q; W_{pad}^Q][W^K; W_{pad}^K]^T K^T$. As $W_{pad}^Q$ and $W_{pad}^K$ are zero matrices, $[W^Q; W_{pad}^Q][W^K; W_{pad}^K]^T = W^Q(W^K)^T$, i.e., $Q_{pad} K_{pad}^T = QK^T$. Therefore, padding does not affect the attention weight matrix.

3. For norm layers, padding a zero matrix may change the scaling factor, e.g., variance in RMSNorm. When computing these scaling factors, ignoring the additional zeros can keep the original results.

Due to the complexity of implementing TP with an existing framework (vLLM (Kwon et al., 2023)), we leave this as a promising future work to integrate TP with PEARL.

## G  EXPERIMENT RESULTS UNDER LIMITED GPU RESOURCES

Although our PEARL parallels the draft model and the target model at the algorithmic level, it still remains a challenge for deployment at the hardware level in the GPU-constrained scenarios, which we refer to as "co-locate" setting or resource competitions (RC). The key problem lies in the nature of GPU hardware design —- two running processes on the same GPU will compete for GPU resources, which leads to significant slowdowns.

However, in real-world LLM applications, the large-scale target model is usually placed with more than 1 GPU to handle more requests and long context inference, while the small-scale draft model only needs 1 GPU for inference. In this case, pipeline parallelism (PP) is the most common solution to serve the target model with multiple GPUs, which distributes the parameters to different GPUs and conducts computations sequentially with these GPUs.

Inspired by this observation, we propose an improved version of PEARL to effectively utilize GPU computation resources with PP without resource competitions. The key idea is to transfer the computation of the draft model to another GPU when the target model is running on a specific GPU. Specifically, we transfer the first $\lceil \frac{\gamma}{2} \rceil$ draft token generation to the last device, while the last $\lfloor \frac{\gamma}{2} \rfloor$

draft tokens are generated with the first device. As the computation of the target model is conducted sequentially with multiple GPUs, this method could effectively utilize the GPU resources to avoid resource competition.

Take an instance of $c = 5$, the target model is placed with $g = 4$ GPUs, we denote the time for a target model forward as $t$, and the time that the target model runs at GPU 0 is $\frac{t}{4}$. To analyze the GPU utilization in detail, we split $t$ into $gc = 20$ steps, where each step $\eta = \frac{t}{20}$. During one target model forward, the occupied GPU number in 20 steps is given by:

$$M_p: \quad 0, 0, 0, 0, 0; \quad 1, 1, 1, 1, 1; \quad 2, 2, 2, 2, 2; \quad 3, 3, 3, 3, 3; \tag{2}$$

Then we can further analyze the occupied GPU number of the draft model with proposed methods. First, as the draft model can generate $c$ tokens in 20 steps, it only needs 4 steps to generate 1 draft token. Taking $\lceil \frac{\gamma}{2} \rceil = 3, \lfloor \frac{\gamma}{2} \rfloor = 2$, the first $3 \times 4 = 12$ steps of the draft model will occupy the GPU 3, while the last $2 \times 4 = 8$ steps of the draft model will occupy the GPU 0. Therefore, the occupied GPU number of the draft model in 20 steps is given by:

$$M_q: \quad 3, 3, 3, 3; \quad 3, 3, 3, 3; \quad 3, 3, 3, 3; \quad 0, 0, 0, 0; \quad 0, 0, 0, 0; \tag{3}$$

In this way, the draft model and the target model will occupy different devices at each step, which effectively avoids resource competition. However, in real-world settings, moving the draft model from the last device to the first device is non-trivial and costly. As a compromise, We propose to load the draft model both at the first device and the last device. During the inference process, we only move the intermediate KV Cache from the last device to the first device. Many KV Cache compression methods can help further reduce the cost. As the draft model is relatively small, the KV cache itself does not incur significant memory overhead. For example, with Llama 3.1 8B, batch size=1 and input length=1024, the size of the KV cache is $2 \times 2 \times 32 \times 1 \times 8 \times 1024 \times 128 \approx 0.13\text{GB}$. With NVlink, the theoretical time cost for transporting the KV cache is $0.13/300 \approx 0.43\text{ms}$, which is significantly lower than the computational time cost. We provide empirical results of the KV cache transport time cost in Table 16.

Table 16: The empirical time cost of transporting KV cache with different input length. The experiments are conducted with Llama 3.1 8B on HumanEval.

| input length | 128 | 256 | 512 | 1024 |
|---|---|---|---|---|
| empirical time cost | 1.3 ms | 1.4 ms | 1.5 ms | 1.6 ms |

To further evaluate the effectiveness of this method, we conduct some experiments in Table 8, 17 and 18. We found that this strategy allows PEARL to retain $89\% \sim 99\%$ of its original performance, demonstrating the effectiveness of our PEARL in such conditions.

Table 17: Performance comparison of Llama 2 and Llama 3.1 models for PEARL on 4 benchmarks with and without RC (resource competitions). Using proposed strategy in the appendix, PEARL can work well in RC settings with only a slight performance decrease ($< 5\%$).

| | Humaneval | GSMBK | MT-bench | MGSM |
|---|---|---|---|---|
| llama 2 | 32.53 | 30.33 | 24.28 | 30.13 |
| llama 2 (RC) | 31.52 (0.97×) | 29.05 (0.96×) | 22.83 (0.94×) | 28.56 (0.95×) |
| llama 3.1 | 33.56 | 32.97 | 32.14 | 35.02 |
| llama 3.1 (RC) | 31.83 (0.95×) | 31.51 (0.96×) | 30.78 (0.96×) | 33.65 (0.96×) |

Table 18: Performance comparison of Llama 2 and Llama 3.1 models for PEARL on MGSM with and without RC. Using proposed strategy in the appendix, PEARL can work well in RC settings with only a slight performance decrease ($< 10\%$).

| | english (GSMBK) | bengali | german | spanish | french | japanese | russian | swahili | tegulu | thai | chinese | average |
|---|---|---|---|---|---|---|---|---|---|---|---|---|
| **llama 2** | 1.00× | 1.00× | 1.00× | 1.00× | 1.00× | 1.00× | 1.00× | 1.00× | 1.00× | 1.00× | 1.00× | 1.00× |
| **llama 2 (RC)** | 0.96× | 0.95× | 0.95× | 0.94× | 0.95× | 0.96× | 0.94× | 0.95× | 0.96× | 0.93× | 0.92× | 0.95× |
| **llama 3.1** | 1.00× | 1.00× | 1.00× | 1.00× | 1.00× | 1.00× | 1.00× | 1.00× | 1.00× | 1.00× | 1.00× | 1.00× |
| **llama 3.1 (RC)** | 0.98× | 0.94× | 0.95× | 0.94× | 0.94× | 0.96× | 0.96× | 0.95× | 0.98× | 0.98× | 1.00× | 0.96× |

