# OpenReview forum: "PEARL: Parallel Speculative Decoding with Adaptive Draft Length"
_ICLR.cc/2025/Conference — ICLR 2025 Poster_

### Official Review · Reviewer_dgRU · 2024-10-26

**Soundness:** 3
**Presentation:** 3
**Contribution:** 3
**Rating:** 3
**Confidence:** 4

**Summary:**

The authors aim to address two challenges: 1. The mutual waiting problem, which arises when the target model becomes idle while waiting for the draft model to generate tokens, and vice versa. The asynchronous execution of the draft and verification phases leads to inefficiencies. 2. Fixed draft model length.

The authors introduced two strategies to solve this issue: 1. Pre-verification: This strategy involves using the target model to verify the first draft token during the drafting phase. By doing this, PEARL can determine whether the drafted token will likely be accepted or rejected.
If the first draft token is verified and accepted, the draft model can generate additional tokens more confidently. Conversely, if the first token is likely rejected, the draft model can generate fewer tokens, thus saving computational resources and time.

2. Post-verification: In this phase, the draft model generates additional draft tokens while the target model verifies the previously drafted tokens. This allows for a more continuous flow of token generation and verification. By enabling the draft model to produce more tokens during the verification phase, PEARL capitalizes on situations where the target model is actively processing the earlier drafts. This strategy ensures that the draft model is not idle while waiting for the target model to complete its verification, thus maximizing throughput.

**Strengths:**

The paper is clearly written and provides several contributions:

1. PEARL allows the drafting and verification phases to occur simultaneously

2. Instead of using a fixed draft length, PEARL adapts the number of draft tokens generated based on the context and complexity of the task. This flexibility ensures that the draft model generates an appropriate number of tokens, reducing unnecessary computations and improving the acceptance rate of tokens by the target model. This adaptability helps to optimize the inference process for different scenarios.

3. PEARL theoretically demonstrates that it can achieve a higher mean number of accepted tokens compared to existing draft-then-verify methods. This means that more of the generated tokens are useful, leading to better performance overall

**Weaknesses:**

The author claims that they can continue doing draft generation while doing the verification. This raises many questions:

1. The latency of verifying a 70B model is about generating 2.5 tokens on the 8B model. Based on the speedup that the authors provided, this parallel approach would not work beyond a lookahead length of 3, which is shown to be suboptimal empirically. Therefore it is not clear how much this post-verify step improves the performance.

2. Some parameters are underspecified in the tokens accepted per second table (Table). It is not clear under what lookahead length are the baseline numbers achieved, or if they are optimal.

3. The authors picked pipeline parallelism for their implementation. However, while this is a convenient setup for solving the "resource contention" challenge, this is an unreasonable setting and introduces much higher latency in the first place. In deploying a 70B target model with a 7B draft model, using tensor parallelism (TP) can reduce the latency of the target model by leveraging more parallelism in each layer. Therefore, this casts doubt on all the speedup that the authors reported as this causes both the baseline and their reported results to be slower than in a TP setup. Also, with TP, the proposed solution to resource contention in Appendix E would not apply and it is not clear whether the authors can show a similar speedup with their algorithm.

4. Furthermore, using PP instead of TP causes a higher activation memory footprint, reducing the effective batchsize the model can accommodate during decoding, effectively reducing the overall throughput.

**Questions:**

My concerns are raised above.

---

> ### Author Response · Authors · 2024-11-15
> **Response to Reviewer dgRU (part 1/3)**
>
> We thank the reviewer for the insightful and valuable comments. We respond to each comment as follows and sincerely hope that our rebuttal could properly address your concerns. If so, we would deeply appreciate it if you could **raise your score** (3: reject). If not, please let us know your further concerns, and we will continue actively responding to your comments and improving our submission.
>
> > 1. The latency of verifying a 70B model is about generating 2.5 tokens on the 8B model. Based on the speedup that the authors provided, this parallel approach would not work beyond a lookahead length of 3, which is shown to be suboptimal empirically. Therefore it is not clear how much this post-verify step improves the performance.
>
> Thank you for bringing up the insightful comment. We answer it in three folds:
>
> 1. **Clarification of Definition of Draft Length and Window Size**: We apologize for any confusion regarding the definition of the draft length. As in Line 48 of the original manuscript, the draft length refers to the number of tokens generated by the draft model in a continuous execution. This definition indicates that theoretically there is no upper limit to the draft length, as the draft model may generate draft tokens as much as possible. However, the mentioned *lookahead length*, i.e., window size in Line 47, is fixed at any step. In our PEARL, we theoretically show in Section 5.4.1 that the optimal window size should be $c$, i.e., the inference speed ratio between the draft model and the target model, which aligns well with the empirical results in Table 5 of the original manuscript.
> 2. **Motivation of the Post-verify Strategy:** the key idea of post-verify is to allow the draft model to generate more than $\gamma$ tokens without waiting for the target model's verification. For example, given that "the latency of verifying a 70B model is about generating 2.5 tokens on the 8B model and we set the window size as 3" and if the optimal draft length is 12, post-verify fully exploits the drafting ability of the small model to continually generate 4 windows of draft tokens without been interrupted by the target model verification. In this case, the de facto draft length is adaptively adjusted to 12.
> 3. **Effectiveness of the Post-verify Strategy:** As illustrated in Figure 2 (b) in the original manuscript, the optimal draft length dramatically changes in different steps. Using a fixed window size to generate draft tokens cannot fit this change, while our post-verify allows the draft model to generate more tokens when the optimal draft length is larger than $\gamma$. Besides, we also performed an ablation study, as shown in Table 4 (for your convenience, we quote Table 4 as Table R1 here). The results also demonstrate that the absence of the post-verify strategy leads to a substantial reduction in the speedup, which further validates the effectiveness of our post-verify strategy.
>
>
>
> Table R1. Ablation study of post-verify strategy on HumanEval and GSM8K datasets. PEARL *w/o post-verify* denotes PEARL without post-verify strategy.
>
> |                         | Humaneval        | Humaneval        | Humaneval         | GSM8K            |
> | ----------------------- | ---------------- | ---------------- | ----------------- | ---------------- |
> |                         | Codellama 7B&34B | Codellama 7B&70B | Deepseek 1.3B&33B | Llama 2 7B&70B   |
> | PEARL *w/o post-verify* | 1.64$\times$     | 2.57$\times$     | 2.37$\times$      | 2.15$\times$     |
> | PEARL                   | **2.35$\times$** | **3.79$\times$** | **3.48$\times$**  | **2.87$\times$** |

---

> ### Author Response · Authors · 2024-11-15
> **Response to Reviewer dgRU (part 2/3)**
>
> > 2. Some parameters are underspecified in the tokens accepted per second table (Table). It is not clear under what lookahead length are the baseline numbers achieved, or if they are optimal.
>
> We appreciate the reviewer's feedback regarding parameter specification in Table 7 (for your convenience, we quote Table 7 as Table R2 here). In the mean accepted tokens experiments, we select the optimal $\gamma$ for each model pair. Below, we provide a detailed comparison across different $\gamma$ values in Table R3. Our reported baseline results align closely with the most optimal parameters.
>
>
>
> Table R2. Comparison of mean average accepted tokens of vanilla SD methods and PEARL. We run experiments to search the optimal $\gamma$ for SD and report their best results.
>
> |       | Codellama 7&34B   | Codellama 7&70B    | Deepseek 1.3&33B  | Deepseek 6.7&33B  |
> | ----- | ----------------- | ------------------ | ----------------- | ----------------- |
> | SD    | 5.27 ($\gamma=6$) | 8.32 ($\gamma=10$) | 7.23 ($\gamma=8$) | 5.69 ($\gamma=6$) |
> | PEARL | **27.95**         | **26.53**          | **29.65**         | **39.90**         |
>
>
>
> Table R3. Optimal $\gamma$ values of SD for each model pair. (Unit: tokens / second)
>
> | codellama 7&34         | codellama 7&70          | deepseek 1.3&33        | deepseek 6.7&33        |
> | ---------------------- | ----------------------- | ---------------------- | ---------------------- |
> | 30.95 ($\gamma=4$)     | 25.92 ($\gamma=8$)      | 37.22 ($\gamma=6$)     | 30.92 ($\gamma=4$)     |
> | 31.85 ($\gamma=5$)     | 26.02 ($\gamma=9$)      | 38.53 ($\gamma=7$)     | 31.74 ($\gamma=5$)     |
> | **33.57 ($\gamma=6$)** | **27.60 ($\gamma=10$)** | **39.52 ($\gamma=8$)** | **33.77 ($\gamma=6$)** |
> | 33.52 ($\gamma=7$)     | 27.23 ($\gamma=11$)     | 39.38 ($\gamma=9$)     | 33.55 ($\gamma=7$)     |
> | 32.79 ($\gamma=8$)     | 26.65 ($\gamma=12$)     | 38.69 ($\gamma=10$)    | 32.97 ($\gamma=8$)     |
>
>
>
> > 3. The authors picked pipeline parallelism for their implementation. However, while this is a convenient setup for solving the "resource contention" challenge, this is an unreasonable setting and introduces much higher latency in the first place. In deploying a 70B target model with a 7B draft model, using tensor parallelism (TP) can reduce the latency of the target model by leveraging more parallelism in each layer. Therefore, this casts doubt on all the speedup that the authors reported as this causes both the baseline and their reported results to be slower than in a TP setup. Also, with TP, the proposed solution to resource contention in Appendix E would not apply and it is not clear whether the authors can show a similar speedup with their algorithm.
>
> Thank you for your insightful feedback. We acknowledge the benefits of TP in reducing inference latency. We clarify how our PEARL adapts to TP setting and emphasize its contributions as follows:
>
> 1. **Our primary application scenarios are those with sufficient resources**, including adequate GPU capacity, to deploy both the large and small models on separate devices. In this situation, enabling tensor parallelism for both the draft model and the target model is normal and brings no significant overhead. For example, deploy the draft model with 1 A100 GPU with TP=1 and the target model with 7 A100 GPUs with TP=7. We emphasize this scenario is very common and promising in the industry. For example, Mooncake uses prefill-decode separation techniques to further improve the serving efficiency [1].
> 2. While our proposed solution in Appendix E can help PEARL to apply in PP settings, **it remains possible to integrate PEARL with both PP and TP.** Take an instance of our toy example in Lines 849 and 857, in the first 12 micro-steps, theoretically, we can enable tensor parallelism for the target model with TP=3 on GPUs 0, 1, and 2; while in the last 8 micro-steps on GPUs 1, 2, 3. However, we acknowledge that the implementation of tensor parallelism in the speculative setting is challenging, even if the vllm team cannot support vanilla speculative decoding with TP [2]. Moreover, many SD methods also do not support TP implementation as well [3]. We believe that constructing a general TP framework for SD methods is essential, and we think this framework deserves its own publication.
> 3. **We emphasize that the main contribution of our work is the discovery of the mutual waiting problem.** This problem is very common, and will even be exacerbated in TP settings. An observation listed in Table R4 shows that the running speed ratio $c$ between the draft model and the target model is even smaller in TP settings, which leads to a more severe mutual waiting problem. We believe this is a fundamental obstacle for existing SD frameworks to deploy in real-world applications, and our PEARL stands as a pioneer in work to eliminate the mutual waiting problem.

---

> ### Author Response · Authors · 2024-11-15
> **Response to Reviewer dgRU (part 3/3)**
>
> Table R4. Inference speed ratio in TensorRT LLM (with TP) and Huggingface (without TP).
>
> | Framework                | batch size | input length | output length | Llama2 7B speed (tok/s) | Llama2 70B speed (tok/s) | speed ratio |
> | :----------------------- | :--------- | :----------- | :------------ | :---------------------- | :----------------------- | ----------- |
> | TRT LLM (with TP)        | 1          | 512          | 512           | 68                      | 31                       | 2.19        |
> | TRT LLM (with TP)        | 1          | 1024         | 1024          | 65                      | 30                       | 2.17        |
> | huggingface (without TP) | 1          | 512          | 512           | 49                      | 10                       | 4.9         |
> | huggingface (without TP) | 1          | 1024         | 1024          | 49                      | 10                       | 4.9         |
>
> [1] https://github.com/kvcache-ai/Mooncake
>
> [2] https://docs.vllm.ai/en/v0.5.5/models/spec_decode.html
>
> [3] https://github.com/dilab-zju/self-speculative-decoding/issues/21
>
>
>
> > 4. Furthermore, using PP instead of TP causes a higher activation memory footprint, reducing the effective batchsize the model can accommodate during decoding, effectively reducing the overall throughput.
>
> Thanks for your constructive comments.
>
> 1. **Clarification of TP and PP.** TP and PP are both effective ways for inference acceleration and can be combined for better efficiency.  Several LLM inference frameworks, such as TRTLLM [5] and VLLM [6], demonstrate successful implementations where TP and PP are used in tandem. In our implementation, we choose PP as basic parallelism due to its wider applications and simpler implementations.
> 2. **The potential of integrating PEARL into the existing TP framework.** We also provide an idea of TP implementation as an alternative parallelism strategy in response to weakness 3. To clarify, the key idea of PEARL is to utilize the underutilized GPU computations due to the mutual waiting problem, and this does not conflict with tensor parallelism. Therefore, our PEARL can be integrated with the existing TP framework and we leave it as a future work.
> 3. **The batchsize issue.** It is important to note that the speculative decoding (SD) framework performs suboptimally in high-batch settings, as outlined in recent findings from the VLLM blog [4]. SD leverages redundant compute capacity, but in high batch-size regimes, the amount of redundancy diminishes, leading to an overhead from token rejection that outweighs the benefits of SD. This issue is exacerbated as the number of rejected tokens increases, resulting in a significant slowdown. In contrast, PEARL, with its adaptive draft length mechanism, **reduces this overhead by adaptively adjusting the draft length**, allowing it to better accommodate larger batch sizes and improving efficiency compared to SD.
>
> [4] https://blog.vllm.ai/2024/10/17/spec-decode.html
>
> [5] https://github.com/NVIDIA/TensorRT-LLM
>
> [6] https://github.com/vllm-project/vllm
>
> We humbly hope our response has addressed your concerns. If you have any additional concerns or comments that we may have missed in our responses, we would be most grateful for any further feedback from you to help us further enhance our work.

---

> ### Comment · Reviewer_dgRU · 2024-11-20
>
> I still have some doubts about the authors' responses.
>
> The author claims, "Our primary application scenarios are those with sufficient resources". However, this does not address the issue at hand. Given any amount of hardware resources, if the intermediate memory cost is not carefully studied or addressed, a workload that used to fit on n GPUs (say 2 or 4) would now require 2n GPUs (say 4 or 8), which would cause a direct reduction in overall throughput. That was the concern raised in the original rebuttal, which is that the authors picked a setting known to be resource-inefficient, and the technique that the authors proposed only works in that specific setting.
>
> Then the authors claim that they chose PP due to simplicity in implementation. This is a very bizarre claim given that PP is known to be difficult to manage and implement due to the complexity associated with layer sharding (Compared to TP/DDP/FSDP, which one can implement with a one-line or few-line wrappers, using PP would require manual sharding of models and carefully written code for placement and deployment. For all the benefits that PP can have, simplicity in implementation is not known to be one of them). However, I will not dive into this further since this is not the topic of the paper.
>
> In addition, the example that the authors gave on TP also would not work in practice since most of the existing frameworks would require layer width to be divisible by TP size (which would not work with TP=3/7 as shown in the author's example). This further validates my concern that the proposed approach only works in the PP setting, which is sub-optimal for deploying SD.
>
> Therefore, my concern that the proposed method only works in a setting that is known to be resource-inefficient still stands and I am inclined to keep my rating.

---

> > ### Author Response · Authors · 2024-11-21
> > **Response to Reviewer dgRU**
> >
> > Thank you for volunteering your time and effort to review our paper. Below, we address your comments point by point.
> >
> > > The author claims, "Our primary application scenarios are those with sufficient resources". However, this does not address the issue at hand. Given any amount of hardware resources, if the intermediate memory cost is not carefully studied or addressed, a workload that used to fit on n GPUs (say 2 or 4) would now require 2n GPUs (say 4 or 8), which would cause a direct reduction in overall throughput. That was the concern raised in the original rebuttal, which is that the authors picked a setting known to be resource-inefficient, and the technique that the authors proposed only works in that specific setting.
> >
> > We apologize for any misunderstanding of our PEARL. However, in our claimed resource-adequate scenarios, PEARL does not require 2n GPUs, as the scale of the draft model is usually much smaller than the target model by an order of magnitude, and **its resource demand is much smaller than n GPUs** (e.g., 1 additional GPU for LLaMA 2 7B). Given the most common situation which deploys a large-scale target model on 8 GPUs for auto-regressive decoding, PEARL utilizes 7 GPUs to deploy the target model, while the last GPU is used to deploy the draft model. Reducing 1 GPU number of the target model does not affect the target model's inference speed, while our PEARL could significantly improve the overall throughput.
> >
> > > Then the authors claim that they chose PP due to simplicity in implementation. This is a very bizarre claim given that PP is known to be difficult to manage and implement due to the complexity associated with layer sharding (Compared to TP/DDP/FSDP, which one can implement with a one-line or few-line wrappers, using PP would require manual sharding of models and carefully written code for placement and deployment. For all the benefits that PP can have, simplicity in implementation is not known to be one of them). However, I will not dive into this further since this is not the topic of the paper.
> >
> > Thanks for your insightful comments. PP together with TP can be integrated into PEARL to achieve better acceleration. In our current implementation, we choose Transformers [1] as our framework, and PP can also be implemented with a one-line code. We provide the implemented code as follows:
> >
> > `target_model = AutoModelForCausalLM.from_pretrained(target_model_path, device_map="auto", torch_dtype=torch.bfloat16).eval()`
> >
> > > In addition, the example that the authors gave on TP also would not work in practice since most of the existing frameworks would require layer width to be divisible by TP size (which would not work with TP=3/7 as shown in the author's example). This further validates my concern that the proposed approach only works in the PP setting, which is sub-optimal for deploying SD.
> >
> > Thanks for your valuable feedback. We answer this question in three-folds:
> >
> > 1. Our PEARL can utilize tensor parallelism for further acceleration in some specific settings. For example, with 5 GPUs, we can deploy the draft model with 1 GPU (TP=1) and the target model with 4 GPUs (TP=4).
> > 2. Meanwhile, we can use padding techniques to address the division issue directly. For example, given a weight matrix $W \in \mathcal{R}^{4096\times 4096}$ and TP=7, we can append an extra zero matrix $W_{pad}\in \mathcal{R}^{4096\times 6}$ to $W$, and form a padded weight matrix $\hat{W}\in \mathcal{R}^{4096\times 4102}$. In this way, the dimension issue can be effectively addressed. For other $W$ and TP size pairs, we can get the padded matrix similarly. Due to a limited discussion period, we are unable to implement them together in PEARL. We are also committed to integrating these techniques into PEARL as a promising future work.
> > 3.  We also deeply understand the reviewer's concerns regarding the practical deployment of the PEARL framework. However, we would like to emphasize that PEARL is proposed as **a new framework aimed at addressing the intrinsic mutual waiting problem inherent in the SD framework**. Just as the SD framework initially could not be deployed on traditional vLLM or TRTLLM frameworks, but later gained effective support from these platforms [2, 3], we believe that, with the help of the community, our PEARL can also be effectively deployed.
> >
> > We humbly hope our response has addressed your concerns. If you have any additional concerns or comments that we may have missed in our responses, we would be most grateful for any further feedback from you to help us further enhance our work.
> >
> > [1] https://huggingface.co/docs/transformers/index
> >
> > [2] https://github.com/NVIDIA/TensorRT-LLM/issues/169
> >
> > [3] https://github.com/vllm-project/vllm/issues/942

---

> > > ### Comment · Reviewer_dgRU · 2024-11-24
> > >
> > > Regarding TP/PP number of workers issue, in most of the existing frameworks, the reality is that you need to double your GPU consumptions due to tensor dimensions having to be divisible by the number of workers. But this is not a major concern so I will not go into details.
> > >
> > > My understanding and the HuggingFace document states that PP is not current supported nor is TP: https://huggingface.co/docs/transformers/v4.13.0/en/parallelism.
> > >
> > > Can you explain further how you implemented this feature?
> > >
> > > Also, conceptually it is not clear how your described solution would work in a TP+PP or even TP only scenario, which would be more memory-efficient as the current solution described in only works in PP.

---

> > > > ### Author Response · Authors · 2024-11-25
> > > >
> > > > We thank the reviewer for the insightful and valuable comments.
> > > >
> > > > Regarding the PP implementation, our code is based on transformers and accelerate, which can achieve basic PP by splitting the model parameters into multiple chunks and iteratively using these chunks for inference: https://huggingface.co/docs/accelerate/usage_guides/big_modeling.
> > > >
> > > > Regarding the TP implementation, we have conceptually proposed a feasible solution in previous response, which is orthogonal to the already implemented version of PP. However, we have also noted that integrating both TP and PP based on Hugging Face's framework is currently quite challenging. We consider this integration as a promising future work.
> > > >
> > > > We humbly hope our response has addressed your concerns. If you have any additional concerns or comments that we may have missed in our responses, we would be most grateful for any further feedback from you to help us further enhance our work.

---

> > > > > ### Comment · Reviewer_dgRU · 2024-11-25
> > > > >
> > > > > Thank the authors for the response. I am maintaining my score since my concerns are not fully addressed. I want to point out that, as the authors admit, their method is impossible to implement in more efficient settings without redoing the whole implementation. Therefore, simply arguing how this could be done is insufficient as the method evaluation is done in a suboptimal setting, and is unclear how results will be affected after moving to a more appropriate and efficient one, let alone that their method needs to be adjusted in a TP setting.

---

> > > > > > ### Author Response · Authors · 2024-11-27
> > > > > >
> > > > > > We sincerely thank Reviewer dgRU for your thoughtful review and effort to review our paper.
> > > > > >
> > > > > > The major concern of Review dgRU lies in how PEARL adapts to TP settings, where we are stuck due to its significant complexity and consider a future work. However, we would like to emphasize that our PEARL is motivated by the universal mutual waiting problem, and the most of our contribution lies at the algorithmic level. We re-summarize our contributions here:
> > > > > >
> > > > > > 1. Our work directly **points at the universal mutual waiting problem**, which exists at the basic of the speculative decoding framework.
> > > > > > 2. We propose a novel inference acceleration framework PEARL, which can effectively **alleviate the mutual waiting problem with parallelism and achieve adaptive draft length.**
> > > > > > 3. We provide **theoretical analysis to demonstrate the effectiveness of our PEARL**, together with its convenience of acceleration without tunning $\gamma$.
> > > > > >
> > > > > > We also believe that with the help of the community, our work can strengthen speculative decoding in more scenarios and enable more acceleration for LLM inference.

---

### Official Review · Reviewer_LqAV · 2024-10-30

**Soundness:** 2
**Presentation:** 2
**Contribution:** 3
**Rating:** 6
**Confidence:** 3

**Summary:**

The paper describes and addresses the problem of "mutual waiting" in Speculative decoding (SD) for LLM inference acceleration, where typically, the target model and draft model are mutually blocked by each other since verification can only occur after drafting is completed and vice-versa (draft-then-verify). The authors propose a novel framework to address this mutual waiting problem called "Parallel Speculative Decoding with Adaptive Draft Length" (PEARL) that coordinates the drafting and verification steps partially in parallel by using “pre-verify” and “post-verify” strategies for verifying the first draft token in advance during drafting and generating more drafts during verification, respectively. This can be thought of as a shift from the “draft-then-verify” sequential paradigm into a more parallelized “draft-and-verify” paradigm. In addition to the “pre-verify” and “post-verify” strategies, the authors explore the use of an adaptive draft length to further reduce the mutual-waiting scenario where suboptimal draft lengths are used, whether too short or too long. Using these proposed strategies, the authors are able to showcase relative speedups on top of vanilla SD on a variety of text generation tasks.

**Strengths:**

- Authors clearly present their ideas and propose their PEARL framework with solid working examples and motivations. In particular, the analysis into the mutually blocked asynchronous execution of drafting and verification as well as optimal draft length per decode step were helpful in understanding the potential headroom for a parallelized framework of SD.
- Experiments with PEARL yield a solid speed up of 1.5x over vanilla speculative decoding on common text generation tasks as well as in comparison to other baselines for the HumanEval code generation task
- The paper examined each of the component strategies of PEARL, pre-verify, post-verify, and adaptive length, independently via ablation studies and analysis to isolate and highlight the relative impacts and relationships between each strategy
- Authors provide code implementation for reproducibility

**Weaknesses:**

- The main paper assumes execution scenarios where there are enough resources to run drafting and verification in parallel (e.g. multiple GPUs). However, in many instances, drafting and verification happen in co-located settings (e.g. single GPU) where it is more resource constrained. While the authors do make brief mentions in the main paper about these resource constrained scenarios, their discussion and strategies to address this are limited to a relatively short section in the appendix with not much details on their experimental results. It would be much more fitting to have this section filled with more detail and part of the main paper. For example, the strategy mentioned in the appendix involved copying the drafter model across multiple chips which incurs greater memory cost as well as potential communication cost in having to transport and sync intermediate attention KV caches.
- Additionally, given the fact that drafting and verification is assumed to occur on separate devices in parallel, it would be good to see a mention into the communication overhead of data transfer (i.e. logits for adjusted sampling during verification rejection) as well as a breakdown into the additional computational and power consumption in practice from added drafting and verification calls that PEARL conducts in comparison to SD
- While a variety of tasks (HumanEval, GSM8K & MGM, MT-bench) and baselines (SD, Ouroboros, Lookahead Decoding, Distillspec, Assisted Generation) were mentioned in the experiments, the variety of baselines were only used for the HumanEval code generation task while the remaining GSM8K & MGM and MT-bench tasks only used auto-regressive (AR) and SD baselines.

**Questions:**

- Is additional communication overhead costs incurred when running drafting and verification on separate accelerators? (e.g. logits transport from 2 devices for rejection sampling/verification)
- There are 5 baselines listed but Table 2 and Table 3 only report Auto-regressive and SPEED?
- What is the assisted generation baseline exactly? Does it adjust draft length depending on the number of tokens accepted in the previous iteration?
- What value of gamma was used for the baselines on each task? Was it fixed according to the optimal gamma values determined for PEARL?
- Why do Table 5 and Table 6 differ for HumanEval with gamma=5 for Llama2 7B&70B (40.72 in Table 5 and 30.34 in Table 6)?
- Is there a cap on draft length given the fixed optimal verification window size? Something like 2x? Large gamma values aren’t only detrimental to drafting phase but also incur additional computational cost in verification (verifying 4 tokens vs. 32 can be a significant difference) even for vanilla SD
- Table 9 should clarify that it’s not inference speed time but is the number of model runs? Perhaps good to report the ratio on the side (PEARL has ?x more model runs than SD)
- Line 034: remove “the” in “the natural language”
- Line 046-047: rewrite to “draft tokens that the original large model (referred as the target model) then verifies in parallel…”
- Line 161: “generating” not “generation”
- Line 199: “that stucks the target model” ? Do you mean “blocks”?
- Line 240 remove “have”
- Line 331 “being” not “been”
- Line 352: speed up ratio relative to baseline auto-regressive?
- Line 410: Pearl without post-verify as Pearl w/o “pre-verify”
- Line 412: “exhibits a more pronounced”
- Line 691: “reject some” not “someone”

---

> ### Author Response · Authors · 2024-11-15
> **Response to Reviewer LqAV (part 1/3)**
>
> We thank the reviewer for the insightful and valuable comments. We respond to each comment as follows and sincerely hope that our rebuttal could properly address your concerns. If so, we would deeply appreciate it if you could **raise your score** (5: marginally below the acceptance threshold). If not, please let us know your further concerns, and we will continue actively responding to your comments and improving our submission.
>
> > 1. The main paper assumes execution scenarios where there are enough resources to run drafting and verification in parallel (e.g. multiple GPUs). However, in many instances, drafting and verification happen in co-located settings (e.g. single GPU) where it is more resource constrained. While the authors do make brief mentions in the main paper about these resource constrained scenarios, their discussion and strategies to address this are limited to a relatively short section in the appendix with not much details on their experimental results. It would be much more fitting to have this section filled with more detail and part of the main paper. For example, the strategy mentioned in the appendix involved copying the drafter model across multiple chips which incurs greater memory cost as well as potential communication cost in having to transport and sync intermediate attention KV caches.
>
> We appreciate the reviewer's insightful comments. **In response, we will expand the discussion of resource-constrained scenarios in the main body of the paper, specifically in Section 4.5 (Case Studies).**  We also revise the PDF accordingly. For other weakness, we answer as follows:
>
> 1. **Clarification of application scenarios.** The core idea of SD series methods is to leverage redundant computational resources for acceleration. Our PEARL is motivated by the mutual waiting problem, which indicates that SD methods cannot fully utilize the computational resources. Based on this idea, the main application scenarios of PEARL focus on multi-GPU settings, where adequate computational resources can be exploited. We also would like to emphasize that multi-GPU applications, such as high-throughput inference scenarios and edge computing, are highly relevant in practical deployments, where sufficient resources can be allocated for parallel drafting and verification. These settings are important for scaling the approach to large, complex models.
> 2. **PEARL in resource-constrained scenarios.** While our experiments are conducted primarily on multi-GPU settings, we acknowledge that resource-constrained scenarios are very common, and we provide an adaptive method to deploy PEARL in such scenarios. Regarding the details mentioned in the appendix, we provide further clarification on the memory and communication costs.
>    1. (**Additional Memory Cost**) In the strategy provided in the appendix, PEARL requires loading only 1 additional copy of the draft model. Given that the scale of the draft model is typically smaller than 7B, this additional memory cost is lower than 14GB GPU memory (e.g., Llama 2 7B with bfloat 16). It can be further reduced to < 1B with EAGLE draft heads.
>    2. (**Additional KV cache Sync Cost**) Our strategy does incur additional KV cache sync costs to transport the KV cache. However, as the draft model is relatively small, the KV cache itself does not incur significant memory overhead. For example, with Llama 3.1 8B, batch size=1 and input length=1024, the size of the kv cache is $2\times 2 \times 32\times 1\times 8\times 1024\times 128 \approx 0.13$ GB. With NVlink, the theoretical time cost for transporting the KV cache is $0.13/300\approx 0.43$ ms, which is significantly lower than the computational time cost. Other techniques such as  KV cache compression/quantization can further reduce communication costs. **Besides, this time cost can be parallelized with the target model verification process.** We provide empirical results of the KV cache transport time cost in Table R1. Note that in our implementation, the kv cache is sequentially transported, leading to a significantly larger time cost. However, this time cost can still be ignored (<5% of total time cost). These results demonstrate the effectiveness of our approach in mitigating resource contention and supporting efficient multi-task execution in resource-constrained environments.
>
> Table R1. The empirical time cost of transporting KV cache with different input length. The experiments are conducted with Llama 3.1 8B on HumanEval.
>
> | input length        | 128    | 256    | 512    | 1024   |
> | ------------------- | ------ | ------ | ------ | ------ |
> | empirical time cost | 1.3 ms | 1.4 ms | 1.5 ms | 1.6 ms |

---

> ### Author Response · Authors · 2024-11-15
> **Response to Reviewer LqAV (part 2/3)**
>
> > 2. Additionally, given the fact that drafting and verification is assumed to occur on separate devices in parallel, it would be good to see a mention into the communication overhead of data transfer (i.e. logits for adjusted sampling during verification rejection) as well as a breakdown into the additional computational and power consumption in practice from added drafting and verification calls that PEARL conducts in comparison to SD.
>
> Thank you for your insightful feedback. We measure the time cost of each component in one PEARL step with different sizes of model pairs in Table R2. For a more distinct comparison, we also provide the measurement of SD in Table R3. As shown in these 2 tables, the communication overhead within PEARL is negligible.
>
> Table R2. The time cost of each component in one PEARL step. The experiments are conducted on HumanEval.
>
> |               | llama 2 7&70b | codellama 7&34b | deepseek 1.3&33b |
> | ------------- | ------------- | --------------- | ---------------- |
> | communication | 0.2 ms        | 0.3 ms          | 0.2 ms           |
> | verify        | 1.7 ms        | 1.6 ms          | 1.7 ms           |
> | draft         | 105.1 ms      | 68.9 ms         | 65.1 ms          |
> | target        | 108.0 ms      | 71.1 ms         | 66.1 ms          |
>
> Table R3. The time cost of each component in one SD step. The experiments are conducted on HumanEval.
>
> |        | llama 2 7&70b | codellama 7&34b | deepseek 1.3&33b |
> | ------ | ------------- | --------------- | ---------------- |
> | verify | 2.7 ms        | 2.0 ms          | 2.1 ms           |
> | draft  | 107.5 ms      | 68.0 ms         | 65.7 ms          |
> | target | 115.4 ms      | 69.1 ms         | 67.4 ms          |
>
> For the additional computations and power consumptions within PEARL, we first measure the computations of a single model forward with thop [1]. For Llama 2 7b, the computation is 6.607 GFLOPS per token, while the computation is 68.713 GFLOPS per token for Llama 2 70b. Combining the results in Table 9, PEARL will take more computations of about 41.75% than SD, as well as the power consumption. However, our experiments demonstrate that PEARL significantly outperforms SD with up to 1.5$\times$ inference acceleration, which takes the duty of these additional computations and power consumptions.
>
> [1] https://github.com/ultralytics/thop
>
> > 3. While a variety of tasks (HumanEval, GSM8K & MGM, MT-bench) and baselines (SD, Ouroboros, Lookahead Decoding, Distillspec, Assisted Generation) were mentioned in the experiments, the variety of baselines were only used for the HumanEval code generation task while the remaining GSM8K & MGM and MT-bench tasks only used auto-regressive (AR) and SD baselines.
>
> We update the experiments to include results for the additional baselines (Lookahead, Ouroboros, Assisted Generation with LLaMA 2 7&70B) on the MT-Bench and MGSM tasks in Tables R4 and R5. These results will be added to Tables 2 and 3.
> Regarding Lookahead and Ouroboros, it is important to note that these methods currently do not support the LLama 3.1 models, as their official codes only support transformers<=4.36.2 [2], but the Llama 3.1 models require transformers>=4.43. For DistillSpec, the source code has not been made publicly available, and therefore, we are unable to include it as a baseline in our experiments in a short period. We will actively try to communicate with the authors of DistillSpec and reproduce its performance in the future for a more detailed comparison.
>
> [2] https://github.com/hao-ai-lab/LookaheadDecoding/tree/main
>
>
>
> Table R4. Comparisons of different SD methods on MT-bench with Llama 2 7&70b. The highest speedups are bolden.
>
> |                   | writing  | roleplay | reasoning | math     | coding   | extraction | stem     | humanities | average  |
> | ----------------- | -------- | -------- | --------- | -------- | -------- | ---------- | -------- | ---------- | -------- |
> | AR                | 1.00     | 1.00     | 1.00      | 1.00     | 1.00     | 1.00       | 1.00     | 1.00       | 1.00     |
> | SD                | 1.70     | 1.73     | 1.96      | 2.00     | 1.93     | 2.14       | 1.87     | 1.81       | 1.89     |
> | Lookahead         | 1.31     | 1.24     | 1.50      | 1.51     | 1.38     | 1.40       | 1.29     | 1.27       | 1.36     |
> | assist generation | 1.41     | 1.40     | 1.39      | 1.64     | 1.74     | 1.92       | 1.57     | 1.47       | 1.55     |
> | ouroboros         | 1.42     | 1.35     | 1.40      | 1.61     | 1.35     | 1.67       | 1.44     | 1.36       | 1.45     |
> | **PEARL**         | **2.40** | **2.45** | **2.85**  | **2.79** | **2.67** | **2.92**   | **2.58** | **2.50**   | **2.64** |

---

> ### Author Response · Authors · 2024-11-15
> **Response to Reviewer LqAV (part 3/3)**
>
> Table R5. Comparisons of different SD methods on MGSM with Llama 2 7&70b. The highest speedups are bolden.
>
> |                     | english (GSM8K) | bengali  | german   | spanish  | french   | japanese | russian  | swahili  | tegulu   | thai     | chinese  | average  |
> | ------------------- | --------------- | -------- | -------- | -------- | -------- | -------- | -------- | -------- | -------- | -------- | -------- | -------- |
> | AR                  | 1.00            | 1.00     | 1.00     | 1.00     | 1.00     | 1.00     | 1.00     | 1.00     | 1.00     | 1.00     | 1.00     | 1.00     |
> | SD                  | 2.48            | 2.69     | 2.77     | 2.64     | 2.71     | 2.71     | 2.72     | 2.81     | 2.65     | 2.71     | 2.78     | 2.70     |
> | Lookahead           | 1.23            | 1.34     | 1.51     | 1.50     | 1.48     | 1.29     | 1.43     | 1.60     | 1.28     | 1.23     | 1.48     | 1.39     |
> | ouroboros           | 1.60            | 1.75     | 1.88     | 1.69     | 1.80     | 1.95     | 1.65     | 1.68     | 2.45     | 1.92     | 1.81     | 1.84     |
> | assisted generation | 1.96            | 1.69     | 1.75     | 1.70     | 1.67     | 2.02     | 1.68     | 1.58     | 3.07     | 2.17     | 1.97     | 1.93     |
> | **PEARL**           | **3.82**        | **3.94** | **4.00** | **3.81** | **3.76** | **3.94** | **3.85** | **4.18** | **4.10** | **3.93** | **4.06** | **3.95** |
>
> > 4. Other questions
>
> Phew! Thanks for your time and detailed feedback! Thanks for your clear effort to think critically about how to improve our paper! We answer them respectively.
>
> > Q1: Is additional communication overhead costs incurred when running drafting and verification on separate accelerators? (e.g. logits transport from 2 devices for rejection sampling/verification)
>
> As shown in Table R2, the additional communication overhead incurred by running drafting and verification on separate accelerators is negligible.
>
> > Q2: There are 5 baselines listed but Table 2 and Table 3 only report Auto-regressive and SPEED?
>
> We conduct additional experiments accordingly in Tables R4 and R5.
>
> > Q3: What is the assisted generation baseline exactly? Does it adjust draft length depending on the number of tokens accepted in the previous iteration?
>
> Assisted generation [3] is an improved speculative decoding algorithm. It initially sets the window size as 5, and increases it by 2 if all draft tokens are accepted, or decreases it by 1 otherwise. In our experiments, this strategy cannot handle the dynamic draft length well, sometimes brings additional overhead, and incurs more severe mutual waiting problems.
>
> > Q4: What value of gamma was used for the baselines on each task? Was it fixed according to the optimal gamma values determined for PEARL?
>
> For vanilla speculative decoding, we search the optimal $\gamma$ and report its best performance. For other baselines, we either use the reported number in their original manuscript or using their official codes with default parameters for reproduction.
>
> > Q5: Why do Table 5 and Table 6 differ for HumanEval with gamma=5 for Llama2 7B&70B (40.72 in Table 5 and 30.34 in Table 6)?
>
> We apologize for any confusion that may have arisen from the discrepancy between Tables 5 and 6. To clarify, the results presented in Table 5 correspond to the **CodeLlama** model, whereas the results in Table 6 correspond to the **Llama2** model.
>
> > Q6: Is there a cap on draft length given the fixed optimal verification window size? Something like 2x? Large gamma values aren’t only detrimental to drafting phase but also incur additional computational cost in verification (verifying 4 tokens vs. 32 can be a significant difference) even for vanilla SD
>
> The draft length corresponds to the maximal number of draft tokens that can be accepted at a specific speculative step. Theoretically, there is no fixed upper bound of the draft length, as the draft model can generate draft tokens as many as possible. In PEARL, the verification of a large draft length is split into many windows, and each window contains a fixed number of draft tokens. This chunked verification is parallelized with the drafting process, hence it does not incur additional verification time cost.
>
> > Q7: Table 9 should clarify that it’s not inference speed time but is the number of model runs? Perhaps good to report the ratio on the side (PEARL has ?x more model runs than SD)
>
> Thanks for your suggestions. We have added the statistics accordingly.
>
> > Other typos in the manuscript.
>
> Thank you for pointing out these typos, we will update them together into the revised PDF.
>
>
> We humbly hope our response has addressed your concerns. If you have any additional concerns or comments that we may have missed in our responses, we would be most grateful for any further feedback from you to help us further enhance our work.
>
> [3] https://huggingface.co/blog/assisted-generation

---

> > ### Comment · Reviewer_LqAV · 2024-11-22
> >
> > Thank you for addressing my questions and concerns so thoroughly. I have updated my score accordingly.

---

> > > ### Author Response · Authors · 2024-11-23
> > >
> > > Dear Reviewer LqAV,
> > >
> > > Thank you for your feedback! We sincerely appreciate your acknowledgement of our submission and are grateful for the kindly assistance you have provided in enhancing the quality of our submission.
> > >
> > > Best,
> > >
> > > Authors

---

### Official Review · Reviewer_WhQi · 2024-11-01

**Soundness:** 4
**Presentation:** 3
**Contribution:** 4
**Rating:** 6
**Confidence:** 5

**Summary:**

This paper presents an interesting speculative decoding (SPD) paradigm, PEARL, that attempts to overlap the drafting and verification stages in the standard SPD framework, thereby mitigating the so-called mutual waiting issues. As a training-free SPD method, PERAL achieves a state-of-the-art acceleration ratio with different pairs of LLMs on different domains by switching between pre-verify and post-verify stages to adjust the draft length dynamically.

**Strengths:**

1. The idea is novel to some extent: while the community has noticed that the drafting stage is the bottleneck of the current SPD system and has proposed works to either dynamically adjust the draft length or decode draft tokens in parallel, attempts to pre-verify and post-verity are innovative to lower the proportion of drafting latency in the SPD process.
2. The illustrations are self-explanatory in Figure 3.
3. PERAL eliminates the need to tune the drafting window size according to Section 4.1, a desirable property for other SPD frameworks.

**Weaknesses:**

1. Missing baselines: while comparison with SPD methods that require training (Medusa, EAGLE, etc.) is not expected, there is still a line of works that focus on training-free SPD, such as Self-Speculative [1], Parallel Decoding [2] and REST [3]. Adding these should be able to strengthen this submission, but please do not focus on this during the discussion stage.

Other than these, I don’t find obvious weaknesses in this submission. Please refer to the question section regarding my main concern.

[1]: Jun Zhang, Jue Wang, Huan Li, Lidan Shou, Ke Chen, Gang Chen, and Sharad Mehrotra. Draft & verify: Lossless large language model acceleration via self-speculative decoding.
[2]: Andrea Santilli, Silvio Severino, Emilian Postolache, Valentino Maiorca, Michele Mancusi, Riccardo Marin, and Emanuele Rodolà. Accelerating transformer inference for translation via parallel decoding.
[3]: Zhenyu He, Zexuan Zhong, Tianle Cai, Jason D. Lee, and Di He. REST: retrieval-based speculative decoding.

**Questions:**

I find the following concerns and would like further clarification from the authors.

The core idea behind SPD is to utilize hardware computation redundancy; therefore, running forward passes on drafter and target models simultaneously has to bring additional latencies for both the drafting and verification stages. I am glad to see the paper presents theoretical and empirical analysis, but none of them discussed this. Profiling the latency overhead brought by overlapping two stages could strengthen this paper. (I noticed Appendix E, an engineering technique to get around the resource competition issues, but still, some analysis is expected; let’s say we don’t have a multi-GPU environment to implement the PP solution.

---

> ### Author Response · Authors · 2024-11-16
> **Response to Reviewer WhQi (part 1/3)**
>
> We thank the reviewer for the insightful and valuable comments. We respond to each comment as follows and sincerely hope that our rebuttal could properly address your concerns. If so, we would deeply appreciate it if you could raise your score. If not, please let us know your further concerns, and we will continue actively responding to your comments and improving our submission.
>
> > 1. Missing baselines: while comparison with SPD methods that require training (Medusa, EAGLE, etc.) is not expected, there is still a line of works that focus on training-free SPD, such as Self-Speculative [1], Parallel Decoding [2] and REST [3]. Adding these should be able to strengthen this submission, but please do not focus on this during the discussion stage.
>
> We appreciate the reviewer's insightful comments. We update the experiments to include results for additional training-free speculative decoding baselines (self-speculative, parallel decoding and REST) on the GSM8K benchmark in Table R1. Note that the results of self-speculative decoding and parallel decoding are copied from a recent work [1]. The result of REST is reproduced with their official code [2] and default parameter settings. PEARL outperforms these training-free methods by a large margin. All of these experiments validate the effectiveness of PEARL.
>
>
>
> Table R1. Comparison between some training-free speculative decoding baselines and our PEARL. We report the inference speedup as the evaluation metric. The experiments are conducted on GSM8K with Llama 2 7&70B.
>
> |         | Self-Speculative Decoding | Parallel Decoding | REST         | PEARL            |
> | ------- | ------------------------- | ----------------- | ------------ | ---------------- |
> | speedup | 1.10$\times$              | 0.97$\times$      | 1.39$\times$ | **2.40**$\times$ |
>
>
>
> [1] Xia, Heming, Yongqi Li, Jun Zhang, Cunxiao Du, and Wenjie Li. "SWIFT: On-the-Fly Self-Speculative Decoding for LLM Inference Acceleration." *arXiv preprint arXiv:2410.06916* (2024).
>
> [2] https://github.com/FasterDecoding/REST

---

> ### Author Response · Authors · 2024-11-16
> **Response to Reviewer WhQi (part 2/3)**
>
> > 2. The core idea behind SPD is to utilize hardware computation redundancy; therefore, running forward passes on drafter and target models simultaneously has to bring additional latencies for both the drafting and verification stages. I am glad to see the paper presents theoretical and empirical analysis, but none of them discussed this. Profiling the latency overhead brought by overlapping two stages could strengthen this paper. (I noticed Appendix E, an engineering technique to get around the resource competition issues, but still, some analysis is expected; let’s say we don’t have a multi-GPU environment to implement the PP solution.
>
> Thanks for your insightful comments. We answer the question in three-fold:
>
> 1. **Clarification of the application scenarios**. As illustrated, the key idea of speculative decoding is to exploit the computation redundancy for acceleration. Based on this idea, we observe the mutual waiting problem, which hinders speculative decoding to fully utilize the redundant computational resources. Therefore, the main application scenarios of PEARL focus on the scenarios with adequate computational resources, where speculative decoding cannot sufficiently use these resources.
>
> 2. **PEARL in resource-adequate scenarios.** In such scenarios, the draft model and the target model can be deployed separately, and simultaneously running the draft model and the target model would not bring additional latencies in both drafting and verification stages. We have conducted experiments to verify this conclusion, by running the draft model and the target model in parallel (PEARL) and sequentially (SD) in Table R2 and R3, where we set all the hyper parameters the same for fair comparison. From the results, we can see that  **running the draft model and the target model in parallel only incurs negligible additional latencies.**
>
> 3. **PEARL in resource-constrained scenarios.** While our experiments are conducted mainly on resource-adequate scenarios, we acknowledge that the resource constrained scenarios are very common, and we make our effort to provide a strategy to exploit the underutilized resources in PP settings. We also provide experiments to demonstrate the effectiveness of PEARL in multi-GPU resource constrained scenarios using proposed strategy in PP implementation in Table R4. However, when the resources are too limited to implement PP, our PEARL will degenerate to vanilla SD. The explanations are as follows:
>
>    1. **pre-verify in single-GPU**: the pre-verify requires the target model to take an additional verification simultaneously during the drafting stage. However, the GPU resources are occupied by the draft model, and the target model remains idle until the draft model finishes its drafting process. As the drafting process has finished, the target model will directly verify these draft tokens. In this situation, the pre-verify strategy degenerates to the drafting phase in vanilla SD.
>    2. **post-verify in single-GPU**: the post-verify requires the draft model to generate additional draft tokens simultaneously during the verification stage. However, the target model occupies all GPU resources in single-GPU setting, and no GPU resources can be utilized for drafting. In this situation, the post-verify strategy degenerates to the verification phase in vanilla SD.
>
>    To summarize, when there is no underutilized computational resources in single-GPU settings,  PEARL will degenerate to vanilla SD, but is still faster than auto-regressive decoding.
>
>    However, we notice that in resource-constrained setting, offloading the draft model to CPU is also possible [3]. Therefore, offloading the draft model to CPU to run drafting and verification in parallel is an alternative and promising solution in such scenario. Other CPU-concentrated drafting methods (e.g. retrieval-based drafting) can also consider adopting PEARL framework for further acceleration. We believe that PEARL can help improve the inference efficiency in most scenarios.
>
> We humbly hope our response has addressed your concerns. If you have any additional concerns or comments that we may have missed in our responses, we would be most grateful for any further feedback from you to help us further enhance our work.
>
> [3] Miao, Xupeng, Gabriele Oliaro, Zhihao Zhang, Xinhao Cheng, Zeyu Wang, Zhengxin Zhang, Rae Ying Yee Wong et al. "Specinfer: Accelerating large language model serving with tree-based speculative inference and verification." In *Proceedings of the 29th ACM International Conference on Architectural Support for Programming Languages and Operating Systems, Volume 3*, pp. 932-949. 2024.

---

> ### Author Response · Authors · 2024-11-16
> **Response to Reviewer WhQi (part 3/3)**
>
> Table R2. The time cost of each component in one PEARL step. The experiments are conducted on HumanEval.
>
> |               | llama 2 7&70b | codellama 7&34b | deepseek 1.3&33b |
> | ------------- | ------------- | --------------- | ---------------- |
> | communication | 0.2 ms        | 0.3 ms          | 0.2 ms           |
> | verify        | 1.7 ms        | 1.6 ms          | 1.7 ms           |
> | draft         | 105.1 ms      | 68.9 ms         | 65.1 ms          |
> | target        | 108.0 ms      | 71.1 ms         | 66.1 ms          |
>
> Table R3. The time cost of each component in one SD step. The experiments are conducted on HumanEval.
>
> |        | llama 2 7&70b | codellama 7&34b | deepseek 1.3&33b |
> | ------ | ------------- | --------------- | ---------------- |
> | verify | 2.7 ms        | 2.0 ms          | 2.1 ms           |
> | draft  | 107.5 ms      | 68.0 ms         | 65.7 ms          |
> | target | 115.4 ms      | 69.1 ms         | 67.4 ms          |
>
> Table R4. Performance comparison of Llama 2 and Llama 3.1 models for PEARL on 4 benchmarks with and without RC (resource competitions). Using proposed strategy in the appendix, PEARL can work well in RC settings with only a slight performance decrease ($\approx 5\%$).
>
> |                | Humaneval                | GSM8K                    | MT-bench                 | MGSM                     |
> | -------------- | ------------------------ | ------------------------ | ------------------------ | ------------------------ |
> | llama 2        | 32.53                    | 30.33                    | 24.28                    | 30.13                    |
> | llama 2 (RC)   | 31.52 (**0.97**$\times$) | 29.05 (**0.96**$\times$) | 22.83 (**0.94**$\times$) | 28.56 (**0.95**$\times$) |
> | llama 3.1      | 33.56                    | 32.97                    | 32.14                    | 35.02                    |
> | llama 3.1 (RC) | 31.83 (**0.95**$\times$) | 31.51 (**0.96**$\times$) | 30.78 (**0.96**$\times$) | 33.65 (**0.96**$\times$) |
>
> Table R5. Performance comparison of Llama 2 and Llama 3.1 models for PEARL on MGSM with and without RC. Using proposed strategy in the appendix, PEARL can work well in RC settings with only a slight performance decrease ($< 10\%$).
>
> |                      | english (GSM8K) | bengali      | german       | spanish      | french       | japanese     | russian      | swahili      | tegulu       | thai         | chinese      | average      |
> | -------------------- | --------------- | ------------ | ------------ | ------------ | ------------ | ------------ | ------------ | ------------ | ------------ | ------------ | ------------ | ------------ |
> | llama 2 7&70b        | 1.00$\times$    | 1.00$\times$ | 1.00$\times$ | 1.00$\times$ | 1.00$\times$ | 1.00$\times$ | 1.00$\times$ | 1.00$\times$ | 1.00$\times$ | 1.00$\times$ | 1.00$\times$ | 1.00$\times$ |
> | llama 2 7&70b (rc)   | 0.96$\times$    | 0.95$\times$ | 0.95$\times$ | 0.94$\times$ | 0.95$\times$ | 0.96$\times$ | 0.94$\times$ | 0.95$\times$ | 0.96$\times$ | 0.93$\times$ | 0.92$\times$ | 0.95$\times$ |
> | llama 3.1 8&70b      | 1.00$\times$    | 1.00$\times$ | 1.00$\times$ | 1.00$\times$ | 1.00$\times$ | 1.00$\times$ | 1.00$\times$ | 1.00$\times$ | 1.00$\times$ | 1.00$\times$ | 1.00$\times$ | 1.00$\times$ |
> | llama 3.1 8&70b (rc) | 0.98$\times$    | 0.94$\times$ | 0.95$\times$ | 0.94$\times$ | 0.94$\times$ | 0.96$\times$ | 0.96$\times$ | 0.95$\times$ | 0.98$\times$ | 0.98$\times$ | 1.00$\times$ | 0.96$\times$ |

---

> > ### Comment · Reviewer_WhQi · 2024-11-20
> >
> > Good work. I think the authors addressed my concerns pretty well. I updated my rating accordingly.

---

> > > ### Author Response · Authors · 2024-11-21
> > >
> > > Dear Reviewer WhQi,
> > >
> > > Thank you for your feedback! We sincerely appreciate your acknowledgement of our submission and are grateful for the kindly assistance you have provided in enhancing the quality of our submission. If you have any remaining concerns, we will actively address and respond to your valuable comments.
> > >
> > > Best,
> > >
> > > Authors

---

### Official Review · Reviewer_htL3 · 2024-11-03

**Soundness:** 3
**Presentation:** 2
**Contribution:** 3
**Rating:** 8
**Confidence:** 3

**Summary:**

This paper aims to accelerate LLM decoding by building on top of the popular Speculative Decoding (SD) algorithm. It addresses a key bottleneck in SD, the mutual waiting problem: the draft model and target model often get stuck waiting for each other because of sequential execution of the two logic with fixed draft lengths.

Unlike traditional SD, the proposed algorithm PEARL (Parallel spEculative decoding with Adaptive dRaft Length) generates variable draft lengths and supports asynchronous execution through two new operations: pre-verify and post-verify. The authors evaluated PEARL across various tasks (e.g., HumanEval, GSM8k), observing significant speedups in performance.

**Strengths:**

* The motivation is clear, with strong supporting evidence, such as in Figure 2.
* Effective visualization of the PEARL algorithm, as shown in Figure 3.
* Reduction in manual parameter tuning for gamma, which previously required significant effort in SD (Section 4.1), even if the estimation remains somewhat approximate.
* Extensive evaluation across various tasks, including code generation, reasoning, and multi-round dialogue.

**Weaknesses:**

* The window size for each chunk (gamma) appears to remain fixed, which means the draft length will still be determined by the multiplication of gamma. While this may be unavoidable in the current algorithm, the abstract and introduction suggest that the authors intend to eliminate the need for gamma entirely.
* Pre-verify with only a single token might not be the most reliable method. See below for a question.
* While Section 5 presents a study on various datasets, the paper does not include a detailed analysis, such as step-by-step profiling or the failure rate in pre-verification.

**Questions:**

* The range of 1.50x to 4.43x represents a significant gap. Is there any analysis explaining the reasons for such large differences?
* Considering that pre-verifying a single token may not be the most accurate method for estimating difficulty, can the authors provide empirical evidence or analysis on how effectively the pre-verification of the first token compares to that of other tokens? For example, could there be potential pitfalls when using a very large gamma?
* Could the authors illustrate a few examples of profiling PEARL on real data, similar to Figure 3, but using actual data?

---

> ### Author Response · Authors · 2024-11-17
> **Response to Reviewer htL3 (part 1/3)**
>
> We thank the reviewer for the insightful and valuable comments. We respond to each comment as follows and sincerely hope that our rebuttal could properly address your concerns. If so, we would deeply appreciate it if you could raise your score. If not, please let us know your further concerns, and we will continue actively responding to your comments and improving our submission.
>
> > 1. The window size for each chunk (gamma) appears to remain fixed, which means the draft length will still be determined by the multiplication of gamma. While this may be unavoidable in the current algorithm, the abstract and introduction suggest that the authors intend to eliminate the need for gamma entirely.
>
> Thanks for your insightful comments. We apologize for any misunderstanding regarding to the draft length.
>
> 1. **Clarification on window size $\gamma$ and draft length.** In our paper, the window size $\gamma$ refers to a fixed constant through every step, while the draft length refers to a dynamic number of tokens generated by the draft model in a continuous execution. Note that this draft length is theoretically unlimited, as the draft model can generate draft tokens as many as possible.
> 2. **PEARL adjust the draft length adaptively.** We acknowledge that the draft length is determined by the multiplication of $\gamma$. Suppose the optimal draft length is $\eta$, our PEARL will adjust the actual draft length to the nearest multiple of $\gamma$, i.e., $k\gamma$ such that $k\gamma \geq \eta$ and $(k-1)\gamma < \eta$. In this way, we call this mechanism as "adaptive draft length".
> 3. **PEARL eliminates the burden of tuning $\gamma$.** We apologize for any confusion regarding to this in our manuscript. In the analysis section, we claim that our PEARL eliminates the burden of **tuning** $\gamma$, i.e., the $\gamma$ can be theoretically derived at $c$, which is the speed ratio between the draft model and the target model. With this theorem, we can eliminate the burden of tuning $\gamma$ on different benchmarks or different models, which is a huge amount of effort to achieve best speedup. **However, the window size $\gamma$ here is still important, and we do not claim to eliminate the need of this parameter entirely.**
>
> > 2. Pre-verify with only a single token might not be the most reliable method. See below for a question.
> >
> > Q: Considering that pre-verifying a single token may not be the most accurate method for estimating difficulty, can the authors provide empirical evidence or analysis on how effectively the pre-verification of the first token compares to that of other tokens? For example, could there be potential pitfalls when using a very large gamma?
>
> We appreciate the reviewer's insightful comments. In response, we discuss the pre-verify strategy as follows:
>
> 1. In our PEARL, the key idea of the pre-verify strategy aims at providing an additional first-token verification of the current draft tokens, and if the first token is accepted, PEARL will switch to the post-verify strategy and allows the draft model to generate more draft tokens. **However, as this pre-verify process is running in parallel to the drafting process, the target model can only verify the first draft token in advance, as it only depends on the logits of the last token in  the prefix.** If no additional guess of the draft tokens is given, only the verification of the first draft token is possible.
> 2. We also appreciate the reviewer's perspective of understanding pre-verify with "estimating difficulty", **which inspires a future work to enhance the performance of pre-verify**. That is to say, we can employ a multi-hierarchy speculative strategy, using a smaller model to guess the tokens of the draft model, and use the target model to pre-verify multiple draft tokens if the guesses match [1]. We also conduct some experiments in Table R1 to measure the failure rate of the pre-verify strategy, i.e., the situation that the first draft token is accepted, but some of the rest draft tokens are rejected. In Table R1, this situation corresponds to the situation that last step is post-verify while next step is pre-verify. We can see that this situation is very common in practice, and improve the performance of pre-verify is a promising future work.
>
> [1] Sun, Hanshi, et al. "Triforce: Lossless acceleration of long sequence generation with hierarchical speculative decoding." arXiv preprint arXiv:2404.11912 (2024).
>
> Table R1. Measurement of the PEARL strategy changes on MT-bench with Llama 2 7&70b. We use an output length of 256 for experiment.
>
> | last-step / next-step | pre-verify | post-verify |
> | --------------------- | ---------- | ----------- |
> | pre-verify            | 1380       | 4830        |
> | post-verify           | 4829       | 4470        |

---

> ### Author Response · Authors · 2024-11-17
> **Response to Reviewer htL3 (part 2/3)**
>
> > 3. While Section 5 presents a study on various datasets, the paper does not include a detailed analysis, such as step-by-step profiling or the failure rate in pre-verification.
>
> Thanks for your constructive comments, and we will add a detailed analysis including step-by-step profiling and the strategy changes analysis in the revised manuscript.
>
>
>
> > 4. The range of 1.50x to 4.43x represents a significant gap. Is there any analysis explaining the reasons for such large differences?
>
> We apologize for any confusion in the abstract. $1.50\times$ is the speedup ratio of PEARL over vanilla speculative decoding, and $4.43\times$ is the speedup ratio of PEARL over auto-regressive decoding.
>
>
>
> > 5. Could the authors illustrate a few examples of profiling PEARL on real data, similar to Figure 3, but using actual data?
>
> We provide a simple step-by-step profiling of PEARL with a real data prompt "x+y = 4z, x*y = 4z^2, express x-y in z" in Table R2. To understand the effectiveness of PEARL, we give some explanations about the whole process.
>
> 1. At step 0, the prompt "x+y = 4z, x*y = 4z^2, express x-y in z" is input to the draft model and the target model simultaneously with strategy pre-verify. Within the left of this explanation, we omit this original prompt for simplicity. The draft model outputs [Great, I'm], while the target model outputs [I]. Then, we will use [I] to verify the first token [Great] in the draft tokens. As it is not the same, we reject the draft tokens, **save a verification stage of the other draft tokens for acceleration**, and the output prefix is [I]. As there exists rejected token, the next strategy is still pre-verify.
> 2. At step 1, the prefix [I] is input, and the draft model outputs ['m glad you] and the target model outputs [']. This time, ['] is accepted by the target model, and the next strategy is tuned to post-verify.
> 3. At step 2, the prefix together with the other draft tokens [I'm glad you] is input. The draft model outputs [are interested in exploring] while the target model outputs [m], i.e., the first draft token [m] is accepted, but the second draft token [glad] is rejected. The target model additionally appends [happy] to the prefix. As there exists rejected token, the next strategy is pre-verify.
> 4. At step 3, the prefix [I'm happy] is input, and the draft model outputs [to help you with this] and the target model outputs [to]. This time, [to] is accepted by the target model, and the next strategy is tuned to post-verify.
> 5. At step 4, the prefix together with the other draft tokens [I'm happy to help you with this] is input. The draft model outputs [equation! However, I] while the target model outputs [help you with], i.e., the first three draft token [help you with] is accepted, but the fourth draft token [you] is rejected. The target model additionally appends [your] to the prefix. As there exists rejected token, the next strategy is pre-verify.
> 6. At step 5, the prefix [I'm happy to help you with your] is input, and the draft model outputs [question! However, I] and the target model outputs [question]. This time, [question] is accepted by the target model, and the next strategy is tuned to post-verify.
> 7. At step 6, the prefix together with the other draft tokens [I'm happy to help you with your question! However, I] is input. The draft model outputs [notice that the equation you] while the target model outputs [! However, I notice]. All the draft tokens are accepted, and the next strategy is still post-verify. **Therefore, we save $\gamma$ times of draft model forward for acceleration.**
> 8. At step 6, the prefix together with the other draft tokens [I'm happy to help you with your question! However, I notice that the equation you] is input. The draft model outputs [provided is not correct] while the target model outputs [that the], i.e., the first two draft token [that the] is accepted, but the third draft token [equation] is rejected. The target model additionally appends [equations] to the prefix. As there exists rejected token, the next strategy is pre-verify.
>
>  We hope this step-by-step profiling example can help readers to understand our PEARL better.
>
> We humbly hope our response has addressed your concerns. If you have any additional concerns or comments that we may have missed in our responses, we would be most grateful for any further feedback from you to help us further enhance our work.

---

> ### Author Response · Authors · 2024-11-17
> **Response to Reviewer htL3 (part 3/3)**
>
> Table R2. Simple step-by-step profiling of PEARL with prompt "x+y = 4z, x*y = 4z^2, express x-y in z". We only report the first 7 steps for simplicity. The prompt is selected from MT-bench, and we use Llama 2 7&70b as our base model pair.
>
> | steps | input prefix                                                 | current mode | draft model output           | target model output | judging reason                                               | output prefix                                                |
> | ----- | ------------------------------------------------------------ | ------------ | ---------------------------- | ------------------- | ------------------------------------------------------------ | ------------------------------------------------------------ |
> | 0     | x+y = 4z, x*y = 4z^2, express x-y in z                       | pre-verify   | Great, I'm                   | I                   | great is not I, hence great is rejected, turn to pre-verify  | I                                                            |
> | 1     | I                                                            | pre-verify   | 'm glad you                  | '                   | ' is accepted, turn to post-verify                           | I'                                                           |
> | 2     | I'm glad you                                                 | post-verify  | are interested in exploring  | m                   | m is accepted, but glad is rejected, turn to pre-verify      | I'm happy                                                    |
> | 3     | I'm happy                                                    | pre-verify   | to help you with this        | to                  | to is accepted, turn to post-verify                          | I'm happy to                                                 |
> | 4     | I'm happy to help you with this                              | post-verify  | equation! However, I         | help you with       | the first 3 tokens are accepted, but this is rejected, turn to pre-verify | I'm happy to help you with your                              |
> | 5     | I'm happy to help you with your                              | pre-verify   | question! However, I         | question            | question is accepted, turn to post-verify                    | I'm happy to help you with your question                     |
> | 6     | I'm happy to help you with your question! However, I         | post-verify  | notice that the equation you | ! However, I notice | all previous draft tokens are accepted, keep post-verify     | I'm happy to help you with your question! However, I notice  |
> | 7     | I'm happy to help you with your question! However, I notice that the equation you | post-verify  | provided is not correct.     | that the            | equation is rejected. turn to pre-verify                     | I'm happy to help you with your question! However, I notice that the equations |

---

> > ### Comment · Reviewer_htL3 · 2024-11-24
> >
> > Thank you to the authors for the detailed response, particularly Table R2. I encourage you to include it in the main text or the appendix. It successfully addressed many of my concerns, so I am raising the score.

---

> > > ### Author Response · Authors · 2024-11-25
> > >
> > > Dear Reviewer htL3,
> > >
> > > Thank you for your feedback! We sincerely appreciate your acknowledgement of our submission and are grateful for the kindly assistance you have provided in enhancing the quality of our submission.
> > >
> > > Best,
> > >
> > > Authors

---

### Meta-Review · Area_Chair_saJF · 2024-12-23

**Metareview:**

This paper makes an observation that the drafting phase and the verification phase of speculative decoding can further be parallelized as the current implementations lead to verification stage being on hold when drafting is being done and vice versa. The proposal in the paper makes sense and the empirical evidence in the case with no resource constraints demonstrates the effectiveness of the method. There were several questions by the reviewers that were successfully addressed. I also think the remaining concerns of Reviewer dgRU about tensor parallelism is not a blocker for publication and can be acknowledged in the paper. After reading the paper, the AC finds that the theoretical claims of the paper are questionable, and even likely wrong.
* Theorem 1 claims that the optimal $\gamma = c$; here \gamma must be an integer whereas c is an arbitrary positive real number so the claim cannot hold true in the general form claimed here.
* With reading the proof, Theorem 2 seems to assume that $c -> \infty$ in which case the same result holds for standard SD as well.
* The conclusion after Theorem 2 is wrong because the discussion is missing the free token accepted for standard SD. In fact, the two quantities should agree when $\gamma \to \infty$.

Despite these issues, the AC tends to recommend the paper for acceptance and believes that the empirical results are sufficient for acceptance. However, **the paper is accepted under the condition that Theorem 1 and Theorem 2 are removed; with their proofs removed from appendix as well** and the approximate derivations are used to shed insight and intuition on where the gains come from. Congratulations to the authors!

**Additional Comments On Reviewer Discussion:**

All but one reviewer recommend the paper to be accepted. The AC finds the concerns of Reviewer dgRU to not be a blocker for publication. However, upon read the paper, the AC finds that the theoretical claims are wrong unfortunately. I think the empirical results alone are sufficient for acceptance.

---

### Decision · Program_Chairs · 2025-01-22

Accept (Poster)